# Molecular and structural basis of the chromatin remodeling activity by *Arabidopsis* DDM1

Akihisa Osakabe [1,2,5] ✉, Yoshimasa Takizawa [3,5], Naoki Horikoshi[3], Suguru Hatazawa [3], Lumi Negishi [3], Shoko Sato [3], Frédéric Berger [4], Tetsuji Kakutani [1] ✉ & Hitoshi Kurumizaka [1,3] ✉

The histone H2A variant H2A.W occupies transposons and thus prevents access to them in *Arabidopsis thaliana*. H2A.W is deposited by the chromatin remodeler DDM1, which also promotes the accessibility of chromatin writers to heterochromatin by an unknown mechanism. To shed light on this question, we solve the cryo-EM structures of nucleosomes containing H2A and H2A.W, and the DDM1-H2A.W nucleosome complex. These structures show that the DNA end flexibility of the H2A nucleosome is higher than that of the H2A.W nucleosome. In the DDM1-H2A.W nucleosome complex, DDM1 binds to the N-terminal tail of H4 and the nucleosomal DNA and increases the DNA end flexibility of H2A.W nucleosomes. Based on these biochemical and structural results, we propose that DDM1 counters the low accessibility caused by nucleosomes containing H2A.W to enable the maintenance of repressive epigenetic marks on transposons and prevent their activity.

Transposons are mobile DNA elements that contribute to genetic evolution in animals and plants[1–11]. The expression of transposons that results in their mobilization (referred to as transposition) can potentially disrupt gene function and threaten the integrity of the host genome. Therefore, transposons are generally silenced by heterochromatin formation with the contributions of repressive epigenetic modifications, such as cytosine methylation within DNA and histone H3K9 and H3K27 methylations[12–14]. In plants, several DNA methyltransferases methylate DNA in CG and non-CG contexts, and DNA methylation provides a feedback loop to recruit H3K9 methyltransferases[15–23], which participate in transposon silencing.

In eukaryotes, paralogs of each core histone H2A, H2B, and H3 have been identified as histone variants[24,25], with characteristic genome distributions, production patterns, and functions[26–33]. In *Arabidopsis*, four classes of H2A variants have been identified, and they occupy particular functional chromatin domains[34]. Among them, H2A.W evolved in land plants and is specifically localized in pericentromeric

heterochromatin[34,35]. H2A.W is distinguished from other H2A variants by its unique C-terminal tail containing the KSPKK motif, which stabilizes heterochromatin via interactions with linker DNA[34,36–38], and it cooperates with H3 lysine 9 dimethylation (H3K9me2) to silence transposons[39].

Chromatin remodeling factors are proteins harboring an ATPase domain, and promote the translocation of nucleosomal DNA by distorting histone-DNA interactions. Consequently, DNA sliding and histone removal/exchange/replacement are facilitated within the nucleosome[40,41]. In *Arabidopsis*, 41 proteins belong to the family of sucrose non-fermenting 2 (Snf2) chromatin remodeling factors[42]. Their functions are associated with various physiological pathways, including the control of flowering, flower development, and resistance to pathogens[43–52]. An Snf2-type chromatin remodeling factor named DECREASE IN DNA METHYLATION 1 (DDM1) reportedly functions in DNA methylation maintenance[53–56] and transposon silencing[57–62] in the *Arabidopsis* genome. DDM1 specifically binds H2A.W and mediates its

[1]Department of Biological Sciences, Graduate School of Science, The University of Tokyo, Tokyo, Japan. [2]PRESTO, Japan Science and Technology Agency, Kawaguchi, Japan. [3]Laboratory of Chromatin Structure and Function, Institute for Quantitative Biosciences, The University of Tokyo, Tokyo, Japan. [4]Gregor Mendel Institute (GMI), Austrian Academy of Sciences, Vienna Biocenter (VBC), Vienna, Austria. [5]These authors contributed equally: Akihisa Osakabe, Yoshimasa Takizawa ✉e-mail: akihisa-osakabe@g.ecc.u-tokyo.ac.jp; tkak@bs.s.u-tokyo.ac.jp; kurumizaka@iqb.u-tokyo.ac.jp

deposition over transposons for silencing[35,63]. However, the synergistic action of H2A.W and H3K9me2 accounts for the silencing of less than half of the transposons that are silenced by DDM1[39], suggesting additional modes of action for DDM1, such as allowing DNA methyltransferases to access heterochromatin[64,65]. Nucleosomal DNA is highly protected against DNA methylation[66,67], and thus the chromatin remodeling activity by DDM1 would be required for DNA methylation in the context of heterochromatin. Other than the deposition of H2A.W, in the absence of further biochemical and structural studies, the mechanisms by which DDM1 functions in transposon silencing have remained unclear.

We now report the cryo-electron microscopy (cryo-EM) structures of nucleosomes containing H2A and H2A.W, and the DDM1-H2A.W nucleosome complex. Our cryo-EM structures revealed the flexible entry/exit DNA regions in the H2A nucleosome, possibly caused by its specific amino acid substitutions in the C-terminal docking domain. By contrast, the entry/exit regions of nucleosomal DNA were tightly associated with histone in the H2A.W nucleosome, suggesting the low accessibility to DNA. The DDM1-H2A.W nucleosome structure revealed that DDM1 contacts the N-terminal tail of H4 and the DNA within the H2A.W nucleosome. Crosslinking mass spectrometry and biochemical analyses suggested that DDM1 contacts the H2A.W-specific C-terminal tail and increases the flexibility of the entry/exit DNA regions in the H2A.W nucleosome, leading it to resemble the features of the H2A nucleosome. Based on these results, we propose that DDM1 counteracts the DNA end stability of the H2A.W nucleosome and enables chromatin modifiers to access the H2A.W nucleosome on transposons for repressive mark deposition, promoting their silencing in heterochromatin.

## Results

### Cryo-EM structures of nucleosomes containing H2A and H2A.W

In wild type plants, H2A.W is enriched in the pericentromeric heterochromatin[34,68]. Previous genomic analyses have shown that a moderate decrease, but not complete loss, of non-CG methylation was observed in *h2a.w* mutant plants[68]. The *h2a.w* mutant plants also exhibited ectopic localization of canonical H2A (hereafter referred to as H2A) in pericentromeric heterochromatin. These findings suggested that ectopically incorporated H2A in heterochromatin complements the function of H2A.W for the maintenance of DNA methylation, and prompted us to compare the structures of nucleosomes containing these H2A variants.

We reconstituted nucleosomes composed of a 169 base-pair DNA fragment with the Widom 601 sequence[69] and H2A or H2A.W, together with *Arabidopsis* histones H2B, H3.1, and H4 (Supplementary Fig. 1). We used H3.1 for the nucleosome reconstitution because it is relatively enriched in heterochromatin compared to H3.3[35,70–72]. We determined the cryo-EM structures of nucleosomes containing H2A and H2A.W (Supplementary Table 1 and Fig. 1). A single-particle cryo-EM workflow was performed on the H2A- and H2A.W-nucleosomes, and both structures of nucleosomes containing H2A and H2A.W were determined at 2.9 Å resolution (Supplementary Figs. 2–4). Interestingly, in the H2A nucleosome, the entry/exit regions of nucleosomal DNA are disordered and only 113 base pairs (bps) of DNA are bound to the histone octamer (Fig. 1a). By contrast, in the H2A.W nucleosome, the entry/exit regions are fully wrapped around the histone octamer, and 145 bps of DNA are visualized (Fig. 1a, b). These results suggest that the entry/exit DNA ends of the H2A nucleosome are flexible, as compared to those of the H2A.W nucleosome.

In the H2A nucleosome, we also found ambiguous cryo-EM density maps of the H2A C-terminal docking domain and the H3 αN helix (Fig. 1c and Supplementary Fig. 4). The histone-DNA contacts at the entry/exit regions of the nucleosomal DNA by the H3 αN helix are known to be supported by interactions with the H2A C-terminal docking domain[73–76] (Fig. 1c, d and Supplementary Fig. 4). The H2A Met109, His113, and Leu115 residues are substituted by Leu117, Asn121,

and Val123 in H2A.W, respectively (Supplementary Fig. 5a). Notably, we observed that the H2A.W Asn121 and Val123 residues contact the H3 Ile112 residue (Fig. 1e). In addition, the H2A.W Leu117 residue, which is not conserved in H2A, contacts the Asn98 residue of the same H2A.W, and may stabilize the docking domain of H2A.W (Fig. 1e and Supplementary Figs. 4 and 5). These residues correspond to Leu108, Gln112, and Val114 in human H2A, and disordered regions of the *Arabidopsis* H2A in the nucleosome are clearly visualized in the human H2A nucleosome (Supplementary Figs. 4 and 5). We thus conclude that *Arabidopsis* H2A might weaken the intra- and inter-histone interactions in the nucleosome, resulting in destabilized interactions between the H3 αN region and the entry/exit nucleosomal DNA in the H2A nucleosome (Fig. 1f). By contrast, H2A.W interacts with the H3 αN and α2 regions, resulting in the tight wrapping of the entry/exit DNA regions around histone octamer in the H2A.W nucleosome.

### Cryo-EM structure of the DDM1-H2A.W nucleosome complex

Previous reports have shown that DDM1 changes the DNA register in nucleosomes and functions to maintain DNA methylation in pericentromeric regions[65]. To investigate the mechanism of DDM1 binding to the nucleosome, we determined the cryo-EM structure of the H2A.W nucleosome complexed with full-length DDM1 (Supplementary Table 1, and Fig. 2a, b). We performed a single-particle cryo-EM workflow on the DDM1-nucleosome complex. Three-dimensional (3D) classifications identified the class of DDM1 bound to the H2A.W nucleosome, and the DDM1-H2A.W nucleosome structure was determined at 4.7 Å resolution, in which the local resolution ranges of DDM1 and nucleosome in the complex are approximately 5.2-6.2 Å and 4.2-6.2 Å, respectively (Supplementary Fig. 6). The central nucleosomal DNA region is located on the dyad axis of the nucleosome, and termed superhelical location (SHL0). The nucleosomal DNA locations are named every 10 base pairs from SHL0, as SHL±1, SHL±2, SHL±3, SHL±4, SHL±5, SHL±6, and SHL±7. In the DDM1-H2A.W nucleosome structure, DDM1 binds the nucleosomal DNA at SHL-2 and SHL+6 by crossing the superhelical DNA gyres (Fig. 2b–d). The nucleosomal DNA at SHL-2 is substantially distorted by DDM1 binding (Fig. 2d and Supplementary Fig. 7), in a similar manner to the nucleosomal DNA distortion induced by the yeast Snf2 nucleosome remodeler[41,77–79] (Fig. 2d and Supplementary Fig. 8). Hence this result suggests that DDM1 interacts with the nucleosome through a widely conserved mechanism for nucleosome remodeling.

### DDM1 increases the flexibility of nucleosomal entry/exit DNA regions

In the DDM1-H2A.W nucleosome complex, only 111 base pairs of DNA are bound to the histone octamer, and the entry/exit regions from SHL-5 to SHL-7 and from SHL+6 to SHL+7 are disordered (Figs. 2 and 3). These results suggest that DDM1 binding to the H2A.W nucleosome causes the nucleosomal DNA ends to become more flexible, as in the *Arabidopsis* H2A nucleosome (Fig. 1a).

To test if DDM1 increases the flexibility of nucleosomal entry/exit DNA regions in solution, we conducted a restriction enzyme susceptibility assay. We used the *Msp*I and *Rsa*I restriction enzymes, which recognize the nucleosomal DNA regions 60–64 and 5–8 bases away from the dyad axis of the nucleosome, respectively (Fig. 4a and Supplementary Fig. 1a). *Msp*I cleaves its target site if the DNA ends become accessible in the nucleosome; however, *Rsa*I cleavage efficiency may be unchanged (Fig. 4a). In the absence of DDM1, both *Rsa*I and *Msp*I poorly digested the nucleosomal DNA, reflecting the tight wrapping of the entry/exit DNA region around histone octamer containing H2A.W (Figs. 1b and 4b–g). By contrast, in the presence of DDM1, the susceptibility of nucleosomal DNA to *Msp*I was enhanced, but not to *Rsa*I (Fig. 4b–g). These results suggest that DDM1 changes the flexibility of the entry/exit regions of nucleosomal DNA without nucleosome disassembly.

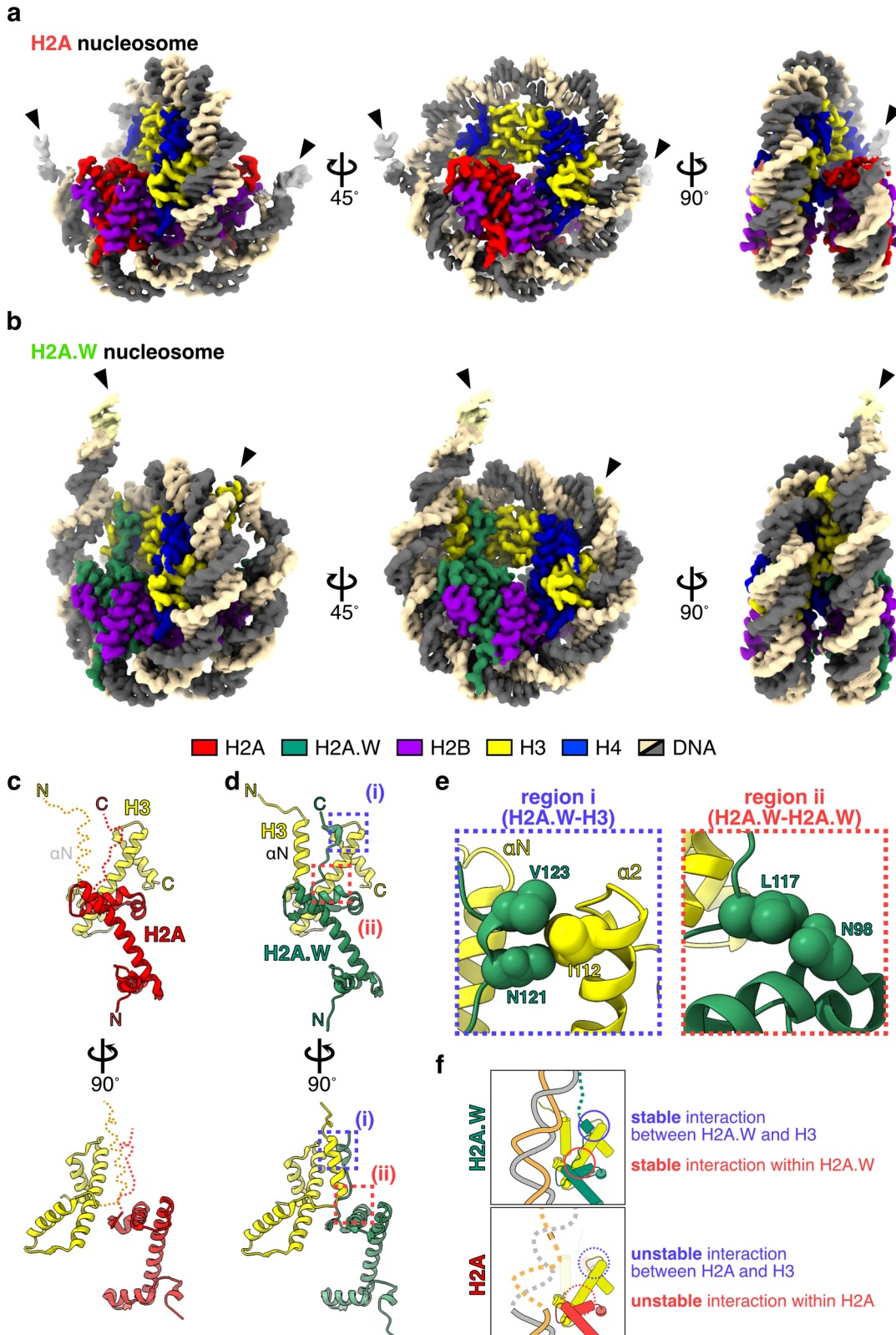

**Fig. 1 | Cryo-EM structures of nucleosomes containing H2A and H2A.W.** Cryo-EM structures of nucleosome containing AtH2A (**a**) and AtH2A.W (**b**). Arrowheads indicate the terminal DNA detected by cryo-EM density maps. Structures of AtH2A (**c**) and AtH2A.W (**d**) complexed with H3. Red and yellow dashed lines indicate the disordered regions in AtH2A nucleosomes (**c**). The two contact sites discussed in panel **e** are enclosed in dashed boxes (i) and (ii). **e** Close-up views of regions (i) and (ii) from panel **d**, where AtH2A.W and H3 are colored green and yellow, respectively. Map-to-density figures of region (i) in the nucleosomes containing AtH2A and AtH2A.W are shown in Supplementary Fig. 4. **f** Graphical summary for the flexibility of the entry/exit nucleosomal DNA ends in the nucleosomes containing H2A.W (upper) and H2A (lower).

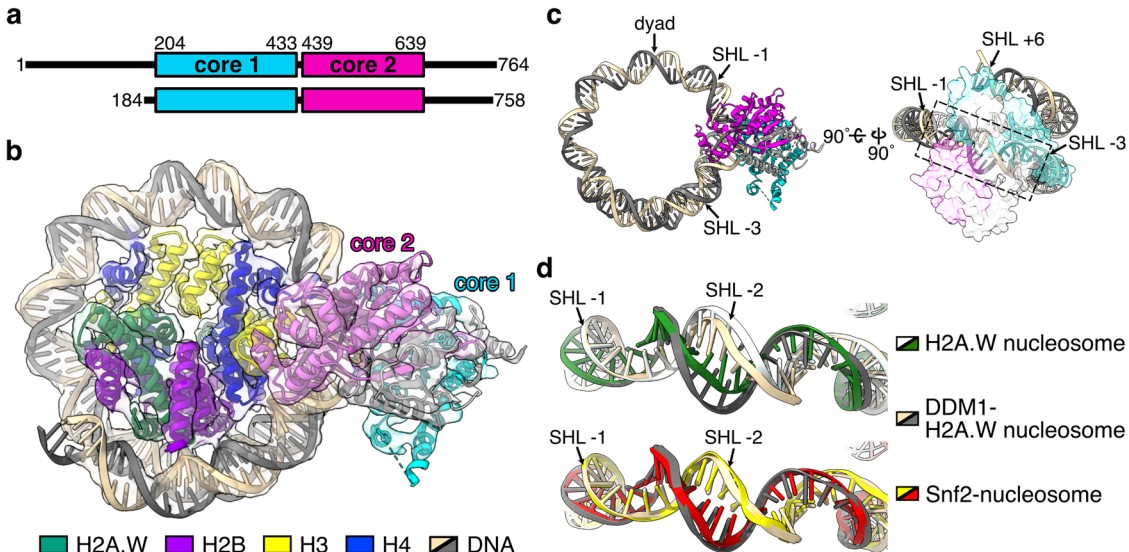

**Fig. 2 | Cryo-EM structure of the AtDDM1-nucleosome complex. a** Schematic representation of the full-length AtDDM1 (upper) and the AtDDM1 fragment observed by cryo-EM (lower). The ATPase core domains 1 and 2 are colored cyan and magenta, respectively. The amino acid sequence alignment between AtDDM1 and *Saccharomyces cerevisiae* Snf2 is presented in Supplementary Fig. 8. **b** Cryo-EM structure of the AtDDM1-nucleosome complex. The atomic structure model of the AtDDM1-nucleosome complex is fitted to the transparent cryo-EM density map. The ATPase core domains 1 and 2 of AtDDM1 are colored cyan and magenta, respectively. **c** Structure of nucleosomal DNA bound by AtDDM1. The dashed box corresponds to the nucleosomal DNA around SHL-2, where DNA distortion was observed. **d** Structural comparison of nucleosomal DNAs bound by AtDDM1 (light orange and gray), ScSnf2 in the absence of ADP (yellow and red, PDB ID: 5X0Y[77] (Snf2-nucleosome complex)), or H2A.W nucleosome (white and green). Map-to-density figures of the nucleosomal DNA around SHL-2 of the H2A.W nucleosome and the DDM1-H2A.W nucleosome complex are shown in Supplementary Fig. 7.

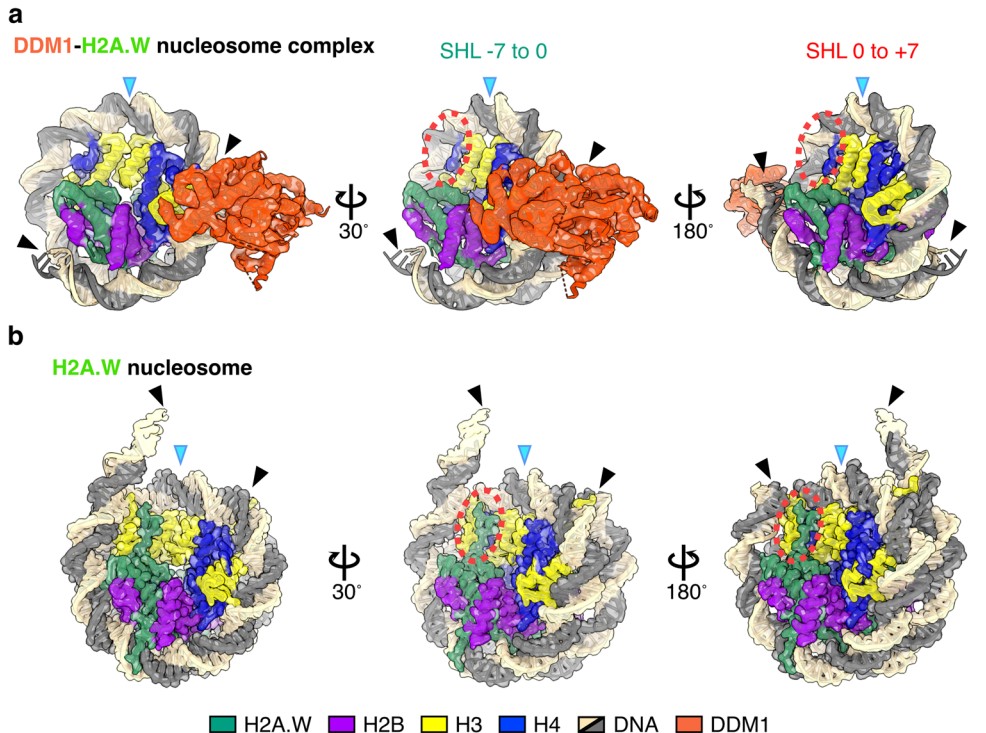

**Fig. 3 | Structural comparison of AtDDM1-bound and AtDDM1-free nucleosomes.** Cryo-EM structures of the AtDDM1-AtH2A.W nucleosome complex (**a**) and the AtH2A.W nucleosome (**b**). Structures of nucleosomal DNA are compared at SHL −7 to 0 (middle) and SHL 0 to +7 (right). Blue arrowheads indicate the dyad axis. Black arrowheads indicate the terminal DNA detected by cryo-EM density maps. Dashed red circles indicate the AtH2A.W docking domain and the H3 αN helix disordered in the AtDDM1-AtH2A.W nucleosome complex.

We further tested the flexibility of the nucleosomal entry/exit DNA regions by fluorescence resonance energy transfer (FRET), using fluorescein and BHQ-1 as its quencher (Fig. 5 and Supplementary Fig. 9). The fluorescein-labeled base and BHQ-1 quencher-labeled base were introduced 57 bases from the 5' end of the forward strand and 16 bases from the 5' end of the complementary strand in 145 base pair DNA fragments (Fig. 5a and Supplementary Fig. 9a). The fluorescein signal was detected with the naked DNA, but was substantially

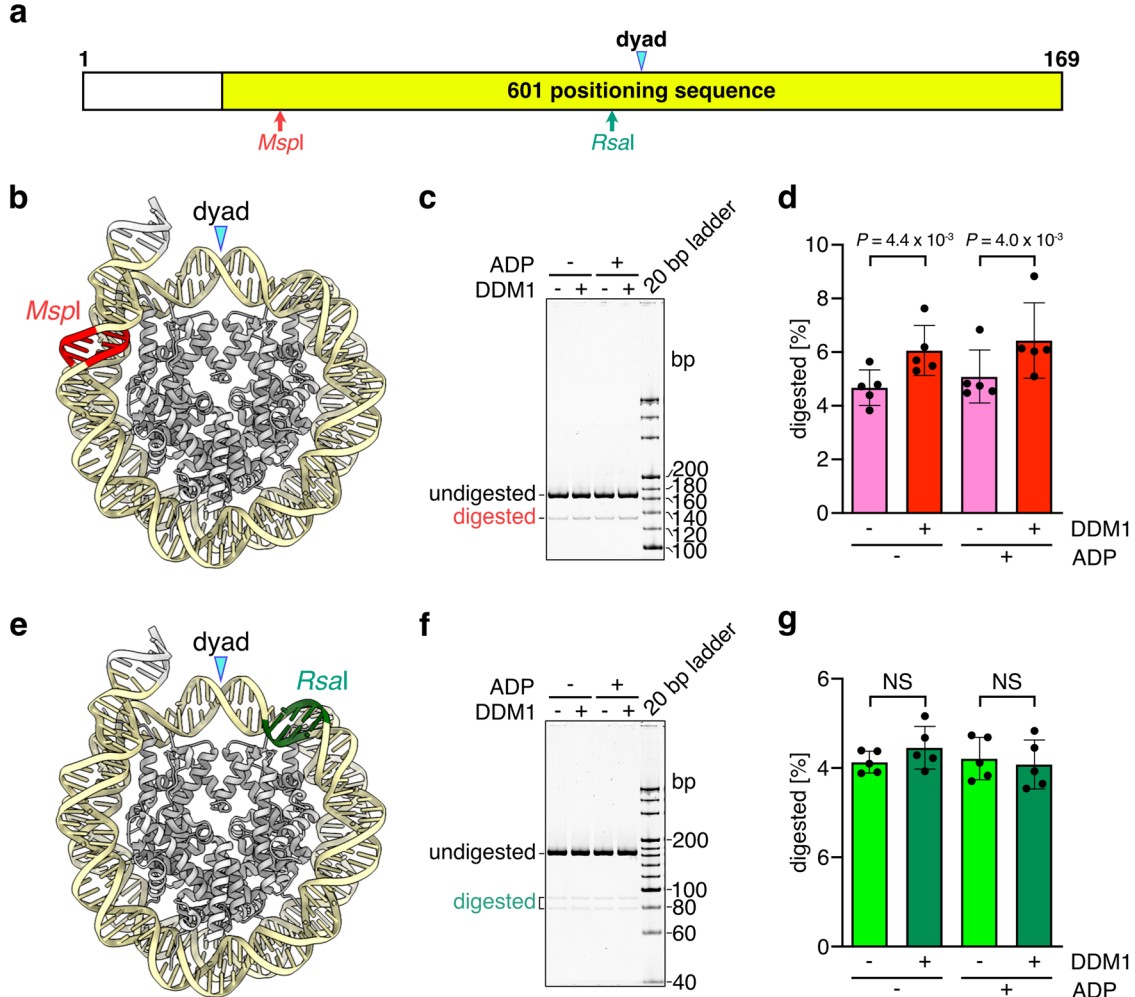

**Fig. 4 | Analyses of the nucleosomal DNA end flexibility by the restriction enzyme susceptibility assay. a** Graphical presentation of the relevant restriction enzyme recognition sites within the DNA fragment used in this experiment. The sequence of the DNA fragment is shown in Supplementary Fig. 1a. Cryo-EM structures of the AtH2A.W nucleosome with the locations of the *Msp*I (**b**) and *Rsa*I (**e**) sites. Native-PAGE analyses of DNA fragments after the restriction enzyme susceptibility assays with *Msp*I (**c**) and *Rsa*I (**f**). Graphical presentations of the restriction enzyme susceptibility assay results with *Msp*I (**d**) and *Rsa*I (**g**). Means and error bars represent SD from five independent experiments. The statistical significance (*P*) was determined using a *t*-test. NS not significant. Source data are provided as a Source Data file.

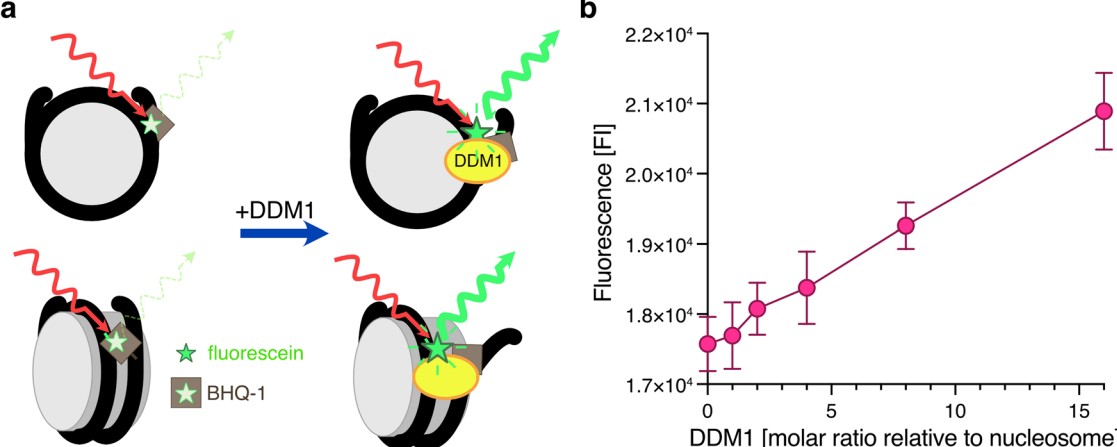

**Fig. 5 | Analyses of nucleosomal DNA end flexibility by the FRET assay.**
**a** Graphical summary of the FRET assay in this study. The emission of fluorescein is inhibited by nucleosome formation because the neighboring BHQ-1 quenches fluorophores, resulting in low fluorescence signals. When the entry/exit nucleosomal DNA ends are unwrapped, fluorescence signals can be detected. **b** Graphical presentation of the FRET assay results with DDM1. Means and error bars represent SD from four independent experiments. Source data are provided as a Source Data file.

suppressed in the nucleosome because of the proximity of BHQ-1 (Supplementary Fig. 9b). The fluorescence signal becomes enhanced when the DNA ends are flexibly disordered, because the BHQ-1 quencher located near a nucleosomal DNA end is farther from the fluorescein in the nucleosome.

We reconstituted the nucleosome containing H2A.W and performed the FRET assay. As expected, the fluorescence signals of the H2A.W nucleosome were drastically enhanced when the nucleosome was disrupted by increasing the NaCl concentration (Supplementary Fig. 9c, d). We found that DDM1 enhanced the fluorescence signal of the H2A.W nucleosome in a concentration-dependent manner (Fig. 5b). This result further supports that DDM1 increases the flexibility of nucleosomal entry/exit DNA regions of H2A.W nucleosome such that it resembles the *Arabidopsis* H2A nucleosome (Fig. 1a).

### Crosslinking mass spectrometric analyses of DDM1-nucleosome complexes

Our biochemical and structural analyses demonstrated that DDM1 binding increased the flexibility of the entry/exit nucleosomal DNA regions of the H2A.W nucleosome. These results suggest that DDM1 binding changes the structure of the H2A.W nucleosome, although we did not detect a direct interaction between H2A.W and DDM1 in our cryo-EM structure of the DDM1-H2A.W nucleosome complex (Figs. 2 and 3). Therefore, we speculated that some disordered regions that were not detected by cryo-EM contribute to the flexible entry/exit DNA ends of the DDM1-bound H2A.W nucleosome.

To identify the specific interactions between DDM1 and H2A.W, we next performed a crosslinking mass spectrometric analysis of the DDM1-nucleosome complex (Fig. 6a and Supplementary Fig. 10). We employed disuccinimidyl suberate (DSS-H12/D12), which crosslinks inter- and intra-molecular lysine residues, as the crosslinker. In our crosslinking mass spectrometric analyses, crosslinked peptides corresponding to the flexible tails of histone proteins and DDM1 were detected (Supplementary Fig. 10). Intriguingly, the DDM1 Lys208 and Lys342 residues were crosslinked with the H2A.W Lys147 and Lys140 residues, which are both located in the flexible C-terminal tail (Fig. 6a). To determine whether the DDM1 Lys208 and Lys342 residues directly interact with the disordered H2A.W Lys147 and Lys140 residues, we present the possible crosslinking areas with colored circles centered on the Cα atom of His113 of H2A.W, with a 115.35 Å radius corresponding to the possible length of the H2A.W 113–140 peptide (Fig. 6b). As shown in Fig. 6b, the DDM1 Lys208 and Lys342 residues are located within this possible crosslinking area, and therefore could directly interact with the H2A.W Lys147 and Lys140 residues in the DDM1-H2A.W nucleosome complex.

Altogether, our cryo-EM structure and crosslinking mass spectrometric analyses suggest that DDM1 opens the entry/exit nucleosomal DNA regions of the H2A.W nucleosome through the interaction with the flexible tails of H2A.W, and this activity possibly promotes the accessibility to DNA in constitutive heterochromatin.

### DDM1 slides nucleosomes regardless of H2A variants

Previous biochemical studies revealed that DDM1 has nucleosome sliding activity[72,80], which is required for the maintenance of DNA methylation in heterochromatin[65]. However, it remained unclear if DDM1 shows a preference for specific histone variants. To clarify this, we performed the nucleosome sliding assay with nucleosomes containing H2A and H2A.W (Fig. 7 and Supplementary Fig. 11). Nucleosome sliding activity can be monitored by migration distance in native-PAGE gels, where faster and slower migrations correspond to the end and middle positions of nucleosomes, respectively (Supplementary Fig. 11b). Under these experimental conditions, the band shift was detected by the addition of DDM1, suggesting that DDM1 has nucleosome sliding activity as previously reported[72,80]. Notably, the sliding activity was detected in both H2A and H2A.W nucleosomes (Fig. 7a, b).

Consistent with the nucleosome sliding activity, DDM1 did not show specific ATPase activity for histone variants, but displayed higher ATPase activity with nucleosomes rather than free DNA (Supplementary Fig. 12).

We further investigated the mechanism by which DDM1 targets and slides nucleosomes. The density of the H4 N-terminal tail was not detected in our cryo-EM structure of the H2A.W nucleosome (Fig. 7c). By contrast, our cryo-EM structure of the DDM1-H2A.W nucleosome complex revealed that the N-terminal tail of H4 is located near the ATPase core domain of DDM1, and is captured by its acidic pocket (Fig. 7c and Supplementary Fig. 13). The binding of the H4 N-terminal tail within the acidic pocket was also reported for the yeast Snf2 protein[41,77–79]. To test the biological significance of the H4 N-terminal tail binding to DDM1, we performed the nucleosome sliding assay with nucleosomes lacking this tail (residues 1–24) (Fig. 7d, e and Supplementary Fig. 14). Consistent with the conserved role of the H4 N-terminal tail in the Snf2-mediated nucleosome remodeling activity[77], the deletion of the H4 N-terminal tail drastically reduced the DDM1-mediated nucleosome sliding (Fig. 7d, e). We thus conclude that DDM1 binds and remodels nucleosomes by a mechanism similar to that described for other chromatin remodelers of the Snf2 family.

## Discussion

In the present study, the cryo-EM structure of the DDM1-H2A.W nucleosome complex revealed that DDM1 binds the nucleosomal DNA at the SHL-2 and SHL+6 positions (Fig. 2). This structure is similar to those reported recently by other groups[72,81]. In addition, a structural comparison between the DDM1-bound H2A.W nucleosome and the DDM1-free H2A.W nucleosome suggested that DDM1 binding caused the H2A.W nucleosome to adopt DNA end flexibility, like the H2A nucleosome (Figs. 2–5). The structural comparison of the H2A and H2A.W nucleosomes suggested that the unique residues of *Arabidopsis* H2A might contribute to the unstable interaction between the docking domain and the H3 αN and α2 helices, resulting in the flexible nucleosomal DNA entry/exit regions in the H2A nucleosome compared to the H2A.W nucleosomes (Fig. 1). Therefore, the DDM1 binding could perturb the histone-DNA contacts around the entry/exit regions of the H2A.W nucleosome. This correlates with the increased accessibility of DNA methyltransferases to DNA and unchanged DNA methylation levels in *h2aw* mutant plants, in which H2A.W is replaced by H2A[68]. The flexibility of the entry/exit nucleosomal DNA does not affect the ATPase activity of DDM1, resulting in the equally efficient sliding of both H2A and H2A.W nucleosomes. These results suggest that DDM1 still functions with the H2A nucleosome for the maintenance of DNA methylation in *h2aw* mutant plants. These findings explain the mechanism by which DDM1 counteracts the low DNA accessibility of H2A.W nucleosomes and enables chromatin modifiers to access heterochromatin (modeled in Fig. 8).

Previous studies have shown that the ATPase activity of the chromatin remodeler ISWI is inhibited by its internal AutoN domain, which is enriched with basic residues and thus resembles the N-terminal tail of H4. This autoinhibition is released via competitive binding with the N-terminal tail of H4[82–85]. Our cryo-EM structures and biochemical analyses demonstrated that the interaction between the N-terminal tail of H4 and DDM1 plays important roles in the DDM1-mediated nucleosome sliding (Fig. 7). We noticed the sequence similarities of conserved regions in angiosperm DDM1 with the AutoN domain of ISWI and the N-terminal tail of H4 (Supplementary Fig. 15a, b). Accordingly, like the ISWI AutoN domain, we propose that the conserved DDM1 region also serves as a negative regulator and function via competitive binding with the H4 N-terminal tail. Note that we did not detect an interaction between the N-terminal tail of H4 and DDM1 in our crosslinking mass spectrometric analyses of the DDM1-nucleosome complex (Fig. 6 and Supplementary Fig. 10). This result might be due to the enrichment of acidic residues and the lack of lysine

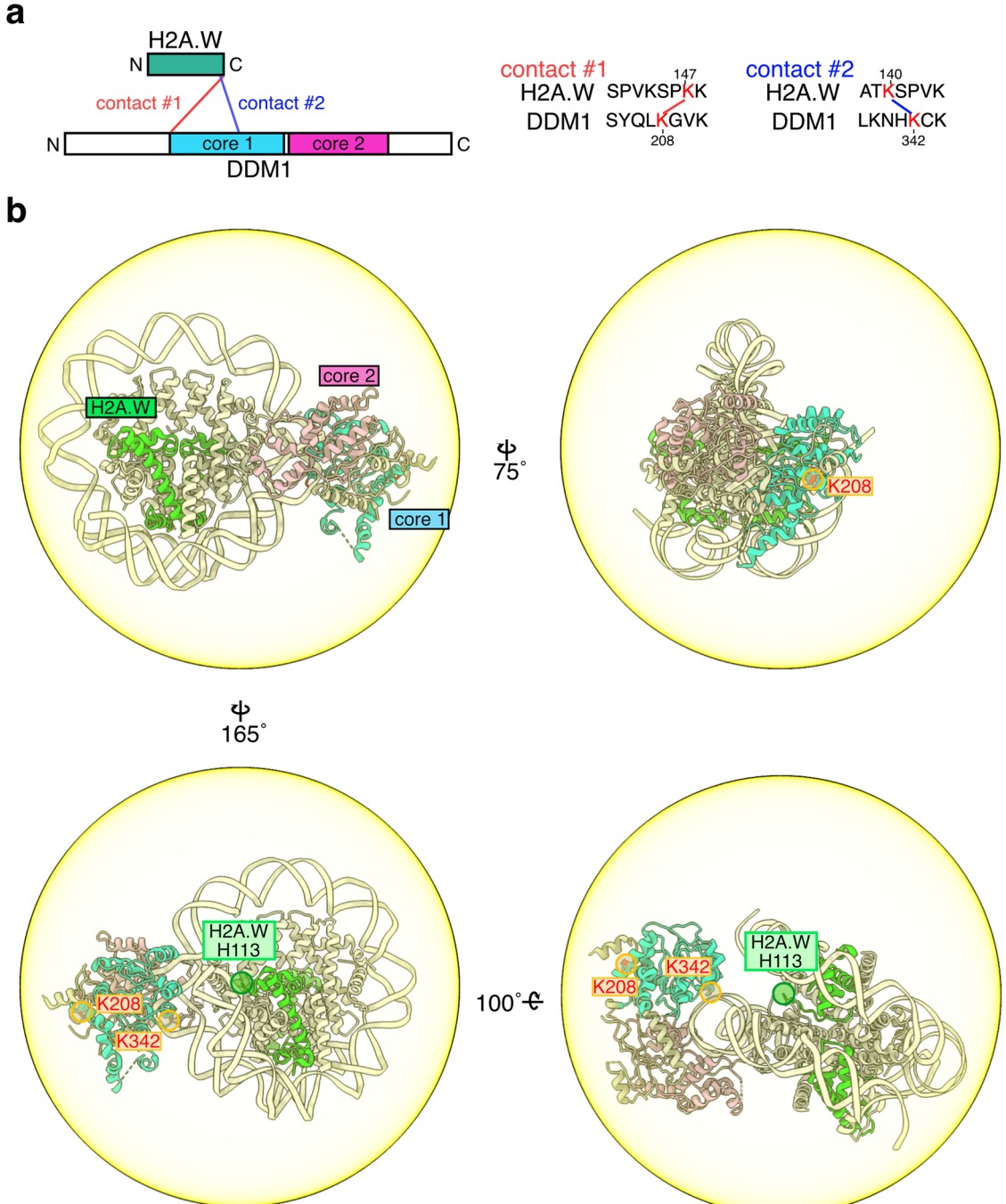

**Fig. 6 | C-terminal tail of AtH2A.W binds DDM1. a** Schematic representation (left) and sequence information (right) for the results obtained by crosslinking mass spectrometry of the DDM1-bound nucleosomes containing AtH2A.W. Two contacts between the H2A.W C-terminal tail and DDM1 are shown. Other crosslinks of the DDM1-bound nucleosomes containing AtH2A.W in the top 25% of ld-scores are shown in Supplementary Fig. 10. **b** Graphical summary of the contacts between the H2A.W C-terminal tail and DDM1, identified by crosslinking mass spectrometry. The yellow circle represents the 115.35 Å radius, corresponding to residues 113–140 of H2A.W (the central point is the Cα atom of His113 of H2A.W), which indicates the possible crosslinking area of H2A.W Lys140 by DSS-H12/D12. The residues of DDM1 contacting the H2A.W C-terminal tail are shown in red with orange circles.

residues within the DDM1 acidic pocket, resulting in the interruption of crosslinking by steric hindrance (Fig. 7c).

Previous genomic analyses have shown the synergistic action of H2A.W and H3K9me2 for transposon silencing[39]. In this study, our crosslinking mass spectrometric analyses identified an interaction between the C-terminal tail of H2A.W and the N-terminal tail of H3 (Supplementary Fig. 10). These regions are located close to the preferred linker DNA for DNA methylation[86], suggesting that the interaction between H2A.W and H3 might contribute to the establishment or

maintenance of the repressive marks for transposon silencing. Further structural and biochemical analyses will be required to clarify this issue.

In addition to our determination of the DDM1 binding mechanism to the H2A.W nucleosome, we also found that DDM1 increases the flexibility of the entry/exit nucleosomal DNAs (Figs. 2–5). This may enhance the accessibility of DNA-binding proteins, including DNA methyltransferases. The binding of the pioneer transcription factor SOX11 to nucleosomal DNA at SHL-2 reportedly leads to a clash with the secondary DNA gyre around SHL+6, inducing the detachment of

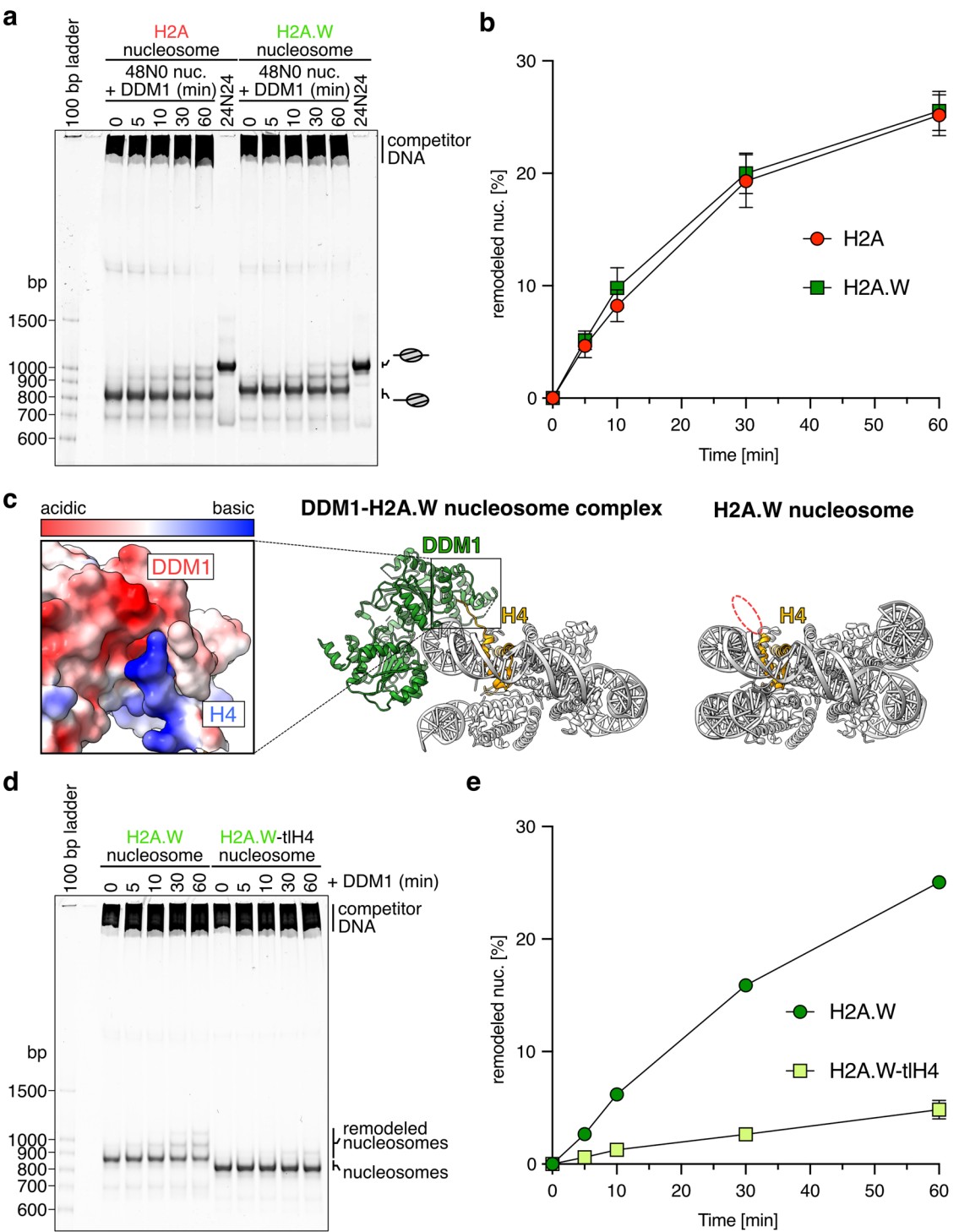

**Fig. 7 | Nucleosome sliding assay. a** Native-PAGE analyses of the nucleosomes containing AtH2A and AtH2A.W after the sliding assay. **b**, **e** Graphical presentation of the nucleosome sliding assay results. Means and error bars represent SD from three independent experiments. The efficiency of the remodeled nucleosome was calculated by the ratio of the intensity of each band and normalized to the ratio obtained at 0 min. Source data are provided as a Source Data file. **c** Overall structures of the AtDDM1-AtH2A.W nucleosome (middle) and the AtH2A.W nucleosome (right). The calculated electrostatic potential of the atomic surfaces of AtDDM1 and the DNAs

the H4 N-terminal tail (residues 19–24) molecules are presented (left). The dashed red circle indicates the disordered regions of the H4 N-terminal tail (residues 19–24) in the AtH2A.W nucleosome. Map to density figures of the H4 N-terminal tail of the AtDDM1-AtH2A.W nucleosome and the AtH2A.W nucleosome are shown in Supplementary Fig. 13. **d** Native-PAGE analyses of the nucleosomes after the sliding assay with nucleosomes containing AtH2A.W, with or without the N-terminal tail of AtH4.

the DNAs around the entry/exit regions of the nucleosome[87]. In contrast to SOX11, DDM1 bound to SHL-2 interacts with the nucleosomal DNA around SHL+6 without steric clashes (Supplementary Fig. 16). DDM1 may directly interact with the H2A.W C-terminal tails including

the docking domains, which are located near the DNA entry/exit regions, and the N-terminal region of H3. These interactions would contribute to the flexible entry/exit nucleosomal DNA ends in the DDM1-H2A.W nucleosome complex.

## wild type

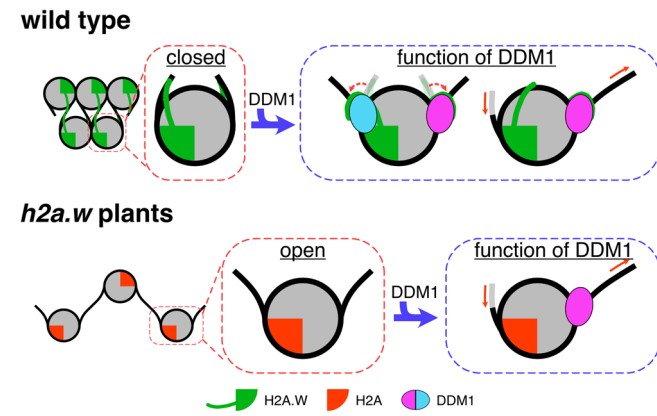

**Fig. 8 | Model of DDM1 activity for the maintenance of repressive marks.** The pericentromeric heterochromatin is occupied with H2A.W, which forms a condensed structure caused by the extended C-terminal tail of H2A.W interacting with the linker DNA, and the H3 αN and α2 helices. In the absence of H2A.W, the pericentromeric heterochromatin is occupied with H2A. The nucleosome containing H2A forms an open structure, caused by the loss of the interactions with the H3 αN and α2 helices. In the presence of DDM1, the C-terminal tail of H2A.W might dissociate from the linker DNA and H3 by interacting with DDM1. In addition, DDM1 slides nucleosomes containing H2A.W and H2A with identical efficiency. These activities might promote the increased accessibility of the heterochromatin to DNA methyltransferases.

The cryo-EM structures of the DDM1-nucleosome (H3.3/H2A.W[72] or H3.1/H2A[81]) complex have been reported. The structure of DDM1 in this study is similar to the published structures except for the loop close to H3, which was disordered in our cryo-EM structure (Supplementary Fig. 17). Intriguingly, we found that the H2A.W docking domain and entry/exit regions of nucleosomal DNA regions were disordered, in contrast to a previous cryo-EM structure (Supplementary Fig. 17). This difference might arise from the use of *Xenopus* H4 in the previous structural analysis. Especially, SWI/SNF-independent (SIN) mutations present in H4 reportedly alter the nucleosome structure[88,89]. Importantly, two amino acid residues, Ile60 and Arg77, are not conserved in *Xenopus* H4. The *Arabidopsis* H4 Ile60 residue exists in the central helix, which could be perturbed in the SIN mutants, and the *Arabidopsis* H4 Arg77 residue is in the loop region located near the nucleosomal DNA backbone. Therefore, it is possible that the structural differences of the nucleosomal DNA ends between the present and previous structures may be due to the use of H4 from different species.

Our structural and biochemical data revealed the mechanism by which the chromatin remodeling factor DDM1 promotes accessibility in constitutive heterochromatin, which is likely crucial for the maintenance of DNA methylation. These results provide important information for future studies of the mammalian DDM1 homolog HELLS/LSH, which mediates the deposition of a histone variant, macroH2A[90,91]. MacroH2A is major component of heterochromatin in mammals[92], and shares sequence similarity and functions with plant H2A.W[38]. HELLS/LSH also controls DNA methylation deposition and transposon silencing[93-95]. HELLS/LSH possesses the basic patch and shares sequence similarities with possible regulatory regions in DDM1 and the AutoN domain of ISWI (Supplementary Fig. 15c). Therefore, it is possible that HELLS/LSH has similar activity to DDM1, thus providing clues about the mechanism of HELLS/LSH and its role in epigenetic human diseases[96].

## Methods
### Preparation and purification of the nucleosome containing Arabidopsis histones
The DNA fragments encoding *Arabidopsis thaliana* AtH2A.13, AtH2A.W.6, AtH3.1, and AtH4 were inserted into the pET-15b vector (Novagen). The DNA fragment encoding AtH2B.9 was inserted into the

pET-15b vector, in which the sequence of the thrombin recognition site (Leu-Val-Pro-Arg-Gly-Ser) was substituted with that of the TEV protease recognition site (Glu-Asn-Leu-Tyr-Phe-Gln-Gly-Ser). The expression and purification of the recombinant *Arabidopsis thaliana* histones were performed by following the methods[37,63,97-99]. Briefly, histones except for H4 were expressed in the BL21(DE3) containing minor tRNA expression vector (Codon (+) RIL) and H4 was expressed in the JM109(DE3) containing minor tRNA expression vector (Codon (+) RIL) as N-terminally hexa-histidine ($His_6$)-tagged proteins. After purification of all histones using nickel-nitrilotriacetic acid agarose (Ni-NTA) resin (QIAGEN) under denaturing conditions, the $His_6$-tag portion was removed by thrombin protease for AtH2A.13, AtH2A.W.6, AtH3.1, and AtH4, and TEV protease for AtH2B.9, respectively. All histones were then purified on a HiPrep SP HP 16/10 cation exchange column (Cytiva) under denaturing conditions with a linear gradient of 200–800 mM NaCl. The DNA fragment encoding truncated AtH4 (aa 25–102) was inserted into pET-15b. Methods for the expression and purification of the truncated AtH4 protein were the same as AtH4. After purification using a cation exchange column, histone proteins were dialyzed against water and freeze-dried. The histone octamers were reconstituted and purified by Superdex 200 gel filtration column (Cytiva) as described[37,98,99]. Briefly, lyophilized histones were mixed at an equal molar ratio and dissolved under denaturing conditions. Then histone octamers were dialyzed against refolding buffer (10 mM Tris-HCl (pH 7.5), 2 M NaCl, 1 mM EDTA, and 5 mM 2-mercaptoethanol). After the dialysis, refolded histone octamers were then purified on a Superdex 200 gel filtration column (Cytiva) under the refolding buffer. The purified histone octamers were frozen in liquid nitrogen and stored at −80 °C until the reconstitution of nucleosomes.

The sequences of the DNA fragments used for nucleosome reconstitution are shown in Supplementary Figs. 1a, 9a, and 11a. The 169 base-pair (24N0) and 193 base-pair (24N24) DNA fragments containing the Widom 601 sequence were purified by polyacrylamide gel electrophoresis (PAGE), using a Prep Cell apparatus as described[100]. Briefly, the DNA fragments were obtained by the excision from the plasmid DNA using EcoRV restriction enzyme and then purified by polyethylene glycol precipitation. The DNA fragments were then further purified by PAGE using a Prep Cell apparatus. The 145 base-pair DNA fragment containing fluorescein and its quencher (BHQ-1) and the 193 base-pair (48N0) DNA fragment containing the Widom 601 sequence were amplified by PCR and purified using a Prep Cell apparatus. The amplification of 145 base-pair DNA fragment containing fluorescein and its quencher (BHQ-1) was performed using primers containing fluorescein or BHQ-1 (FASMAC) at the position described in Supplementary Fig. 9a. The nucleosomes composed of DNA and histone octamers were reconstituted by the salt dialysis method, and then purified by PAGE using a Prep Cell apparatus, as described[37,98,99]. Briefly, the DNA fragments and histone octamers were mixed, and the nucleosomes were reconstituted by the salt dialysis method. The resulting nucleosomes were then purified using a Prep Cell apparatus. The buffer of purified nucleosomes was then exchanged with the storage buffer (20 mM Tris-HCl (pH 7.5), 5% glycerol, and 1 mM DTT) for storage at −80 °C.

### Purification of recombinant Arabidopsis DDM1
The DNA fragment encoding AtDDM1 was inserted into the pET-15b vector (Novagen), in which the sequence of the thrombin recognition site (Leu-Val-Pro-Arg-Gly-Ser-His) was substituted with that of the human rhinovirus (HRV) 3C protease (Leu-Glu-Val-Leu-Phe-Gln-Gly-Pro), and a SUMO tag was integrated between the $His_6$-tag and the HRV 3C protease recognition site. Expression and purification of AtDDM1 were performed by the method described previously[63]. Briefly, AtDDM1 protein was expressed in the BL21(DE3) containing minor tRNA expression vector (Codon (+) RIL) as an N-terminally $His_6$-tagged protein. After purification using Ni-NTA resin, GST-tagged HRV 3C

protease (0.1 mg/mg of His$_6$-SUMO-AtDDM1) was added to remove His$_6$-tagged SUMO from the DDM1 portion. AtDDM1 was subjected to a RESOURCE Q anion exchange column (Cytiva) and collected from the flow-through fractions, and then purified on a RESOURCE S cation exchange column (Cytiva). AtDDM1 protein was further purified on a Superdex 200 gel filtration column using the DDM1 storage buffer (20 mM Tris-HCl (pH 7.5), 150 mM NaCl, 10% glycerol, and 2 mM 2-mercaptoethanol).

### Preparation of nucleosomes and DDM1-nucleosome complexes for cryo-EM

The nucleosomes containing AtH2A or AtH2A.W (150 μg) were cross-linked and purified by the GraFix method[101], using a gradient prepared with buffer A (10 mM HEPES-NaOH (pH 7.5), 1 mM DTT, and 5% (w/v) sucrose) and buffer B (10 mM HEPES-NaOH (pH 7.5), 1 mM DTT, 20% (w/v) sucrose, and 0.1% glutaraldehyde). The nucleosome containing AtH2A.W (2.39 μM) was mixed with the full-length AtDDM1 (14.34 μM) in a total volume of 300 μl reaction buffer (20 mM HEPES-NaOH (pH 7.5), 14.7 mM Tris-HCl (pH 7.5), 150 mM NaCl, 3 mM MgCl$_2$, 1.2 mM DTT, 1 mM ADP, 6.2% glycerol, and 1 mM 2-mercaptoethanol), and incubated at 30 °C for 30 min. Afterwards, the sample was crosslinked and fractionated by the GraFix method, using a gradient prepared with buffer A containing 150 mM NaCl and buffer B containing 150 mM NaCl. The samples were applied to the top of the gradient solution and then centrifuged at 4 °C for 16 h at 125,000 × $g$, using an SW 41 Ti rotor (Beckman Coulter). After centrifugation, aliquots were collected from the top of the solution and analyzed by 6% (nucleosomes) or 4% (AtDDM1-nucleosome complex) non-denaturing PAGE in 0.5 × TBE (44.5 mM Tris-Borate (pH 8.3) and 1 mM EDTA), followed by ethidium bromide staining. The fractions containing the nucleosomes or AtDDM1-nucleosome complex were collected and then desalted on a PD-10 column (Cytiva) by elution with elution buffer (10 mM HEPES-NaOH (pH 7.5) and 2 mM TCEP). The eluted samples were concentrated using an Amicon Ultra-2 centrifugal filter unit (Merck) and stored on ice.

### Preparation of grids for cryo-EM

For the AtDDM1-nucleosome complex (0.25 mg/ml), the nucleosome containing AtH2A (2.0 mg/ml), and the nucleosome containing AtH2A.W (2.0 mg/ml), 2.5 μl portions of samples were applied onto freshly glow-discharged Quantifoil R1.2/1.3, Cu, 200-mesh grids. The grids were blotted for 8 sec at 4 °C in 100% humidity, and then plunge-frozen in liquid ethane by using a Vitrobot Mark IV (Thermo Fisher Scientific).

### Cryo-EM data collection

The AtDDM1-nucleosome complex, the nucleosome containing AtH2A, and the nucleosome containing AtH2A.W were imaged on a Krios G4 microscope (Thermo Fisher Scientific), operated at 300 kV and equipped with a BioQuantum energy filter and a K3 direct electron detector (Gatan) with a slit width of 20 eV, operated in the counting mode at a calibrated pixel size of 1.06 Å. Images of the AtDDM1-nucleosome complex, the nucleosome containing AtH2A, and the nucleosome containing AtH2A.W were recorded at a frame rate of 150 ms for 4.5 s. A nominal defocus range of −1 to −2.5 μm was employed, and the movies were automatically acquired using the EPU software (Thermo Fisher Scientific).

### Image processing

The frames of the movies for the AtDDM1-nucleosome complex, the nucleosome containing AtH2A, and the nucleosome containing AtH2A.W were subjected to motion correction using MOTIONCOR2, with dose weighting[102]. The contrast transfer function (CTF) was estimated using CTFFIND4[103], and RELION4[104] was used for the following image processing. For the AtDDM1-nucleosome complex, a total of 7,004,935 particles from dataset1 and 2,125,419 particles from dataset2 were picked by template-based auto-picking, using the 2D class averages of auto-picked particles based on a Laplacian-of-Gaussian filter as templates, followed by a few rounds of 2D classification to remove junk particles, resulting in the selection of 895,881 and 898,969 particles, respectively. The two datasets were combined, and a de novo initial model generated by Relion4 was low-pass filtered to 60 Å and used as the initial model for the 3D classification. The 3D class with the density map of AtDDM1 containing 128,471 particles was selected, and subjected to focused 3D classification without alignment using the AtDDM1 mask. Subsequently, 34,559 particles selected from the best classes were subjected to Bayesian polishing and CTF refinement. The final postprocessing yielded a cryo-EM map of the AtDDM1-nucleosome complex with a global resolution of 4.71 Å, with the gold standard Fourier Shell Correlation (FSC = 0.143) criteria[105]. The cryo-EM map of the AtDDM1-nucleosome complex was post-processed with the DeepEMhancer software[106].

For the AtH2A and AtH2A.W nucleosomes, 3,651,924 particles and 4,254,793 particles were picked by template-based auto-picking, respectively, using the 2D class averages of auto-picked particles based on a Laplacian-of-Gaussian filter as templates. After 2D classification to remove junk particles, 2,141,126 and 2,579,646 particles were selected for the nucleosomes containing AtH2A and AtH2A.W, respectively. The crystal structure of the nucleosome[107] (PDB ID: 3LZ0 (Nucleosome core particle composed of the Widom 601 DNA sequence)) was low-pass filtered to 60 Å and used as the initial model for the 3D classification. The selected particles were subjected to 3D classification. Subsequently, the best classes from the 3D classifications of the AtH2A and AtH2A.W nucleosomes, containing 411,432 and 196,430 particles, respectively, were subjected to Bayesian polishing and CTF refinement. The final postprocessing yielded cryo-EM maps of the AtH2A and AtH2A.W nucleosomes with global resolutions of 2.94 Å and 2.94 Å, respectively, with the gold standard Fourier Shell Correlation (FSC = 0.143) criteria[105]. The cryo-EM map of the nucleosome containing AtH2A.W was post-processed with the DeepEMhancer software[106].

The local resolutions of the AtDDM1-nucleosome complex, the nucleosome containing AtH2A, and the nucleosome containing AtH2A.W were calculated by RELION-4. Visualization and rendering of all cryo-EM maps were performed with UCSF ChimeraX[108].

### Model building and refinement

Model building was performed with COOT[109], using the crystal structure of the nucleosome[107] (PDB ID: 3LZ0 (Nucleosome core particle composed of the Widom 601 DNA sequence)) and the AtDDM1 structure generated by AlphaFold2[110]. The nucleosomal DNA was automatically fitted into the vacant volume with ISOLDE[111]. The structural models of the AtDDM1-nucleosome complex and the nucleosome containing AtH2A.W were refined by real-space refinement in Phenix[112,113], and validation was performed with MolProbity[114]. The data collection and statistics for the 3D reconstruction and model refinement are shown in Supplementary Table 1.

### Restriction enzyme susceptibility assay

The nucleosomes (0.2 μM) were mixed with DDM1 (1.6 μM) in a total volume of 10 μl reaction solution, containing 20 mM HEPES-NaOH (pH 7.5), 12 mM Tris-HCl (pH 7.5), 150 mM NaCl, 5% glycerol, 5 mM MgCl$_2$, 1.2 mM DTT, 0.8 mM 2-mercaptoethanol, 0.1 mg/ml BSA, and 0 or 1 mM ADP. The restriction enzyme MspI (10 units) or RsaI (10 units) was added to the mixtures and incubated at 30 °C for 60 min. After restriction enzyme digestion, the reaction was terminated by the addition of 5 μl deproteinization solution (20 mM Tris-HCl (pH 8.0), 20 mM EDTA, 0.1% SDS, and 0.5 mg/ml Proteinase K). The resulting DNA was extracted with phenol-chloroform and then analyzed by 10% non-denaturing PAGE in 0.5× TBE. The gel was stained with SYBR Green I solution and DNA was visualized by iBright Imaging Systems.

## FRET assay

The nucleosome (0.05 μM), composed of histones and the DNA fragment containing fluorescein and BHQ-1, was incubated with different concentrations of DDM1 (0, 0.05, 0.1, 0.2, 0.4, and 0.8 μM). The reaction was performed in a 20 μL total volume, containing 23 mM Tris-HCl (pH 7.5), 75 mM NaCl, 1.2 mM DTT, 1 mM 2-mercaptoethanol, 6% glycerol, 0.03% NP-40, and 3 mM MgCl$_2$, and was incubated at 30 °C for 30 min within a 384-well microplate (Greiner Bio-One). After the incubation, the fluorescence intensity of each sample was measured using a Synergy H1M2F (BioTek), with the excitation and emission wavelengths set to 467 nm and 528 nm, respectively.

## Crosslinking mass spectrometry

Nucleosomes (0.2 μM) were mixed with AtDDM1 (0.4 μM) in reaction buffer (20 mM HEPES-NaOH (pH 7.5), 60 mM NaCl, 1 mM MgCl$_2$, 1.1 mM DTT, 1 mM ADP, 4.5% glycerol, 0.03% NP-40, and 0.8 mM 2-mercaptoethanol), and incubated at 25 °C for 30 min. The samples were then crosslinked with 1.6 mM DSS-H12/D12 (Creative Molecules) at 25 °C for 30 min, and the reaction was quenched by the addition of 50 mM Tris-HCl (pH 7.5) followed by an incubation at 25 °C for 15 min. Crosslinking mass spectrometry was performed as described previously[115,116]. Briefly, the samples were dried and dissolved in an 8 M urea solution to a final protein concentration of 1.0 mg/ml. Redissolved samples were reduced by 2.5 mM TCEP, followed by alkylation with 5 mM iodoacetamide. For tryptic digestion, the samples were diluted with a 50 mM ammonium bicarbonate solution to a final concentration of 1 M urea, and then sequencing-grade endopeptidase Trypsin/Lys-C Mix (Promega) was added at an enzyme–substrate ratio of 1:50 wt/wt. The digested samples were applied to a Superdex 30 Increase 3.2/300 (GE Healthcare) column, using buffer containing 25% acetonitrile and 0.1% TFA. The eluted fractions (100 μl) were collected and dried completely. The residues were dissolved in 0.1% TFA and analyzed by liquid chromatography tandem mass spectrometry (LC-MS/MS), using an Orbitrap Fusion mass spectrometer equipped with an Ultimate3000 nano-HPLC system (Thermo Fisher Scientific). The crosslinked peptides were identified using the xQuest/xProphet software (version 2.1.5)[115], and the following criteria were applied to the xProphet results filter: maximum border of MS1 tolerance = 7 ppm, minimum border of MS1 tolerance = −4 ppm, false discovery rate (FDR) <0.05, minimum d-score = 0.95. The crosslinks listed in the top 25% of ld-scores were visualized using the webserver xVis[117]. The LC-MS/MS was performed in 2 technical replicates.

## Nucleosome sliding assay

The nucleosomes (0.22 μM) were mixed with DDM1 (1.77 μM), in a reaction solution containing 13.3 mM Tris-HCl (pH 7.5), 67 mM NaCl, 0.2 mM DTT, 0.9 mM 2-mercaptoethanol, and 5.6% glycerol, and incubated at 30 °C for 15 min. The reaction was then initiated by the addition of ATP, in a reaction solution containing 20 mM HEPES-NaOH (pH 7.5), 12 mM Tris-HCl (pH 7.5), 75 mM NaCl, 0.1 mg/ml BSA, 2.5 mM MgCl$_2$, 1.4 mM DTT, 0.8 mM 2-mercaptoethanol, 5% glycerol, and 1 mM ATP, and incubated at 30 °C for 5, 10, 30, and 60 min. The reaction was stopped by the addition of pUC19 plasmid DNA (0.042 μM) in the presence of 15 mM EDTA. The samples were analyzed by 6% non-denaturing PAGE in 0.5× TBE. The gel was stained with SYBR Green I solution, and the DNA was visualized by an iBright Imaging System (Thermo Fisher Scientific). The quantification was performed with the iBright Analysis Software (Thermo Fisher Scientific). The efficiency of the remodeled nucleosome was calculated by the ratio of the intensity of each band and normalized to the ratio obtained at 0 min.

## DDM1-nucleosome binding assay

The 48N0 nucleosomes (0.2 μM) were mixed with AtDDM1 (0, 0.1, 0.2, 0.4, 0.8, and 1.6 μM) in a total volume of 10 μl reaction buffer, containing 20 mM HEPES-NaOH (pH 7.5), 12 mM Tris-HCl (pH 7.5), 60 mM NaCl, 0.5% glycerol, 1 mM MgCl$_2$, 0.03% NP-40, 1.2 mM DTT, 0.8 mM 2-mercaptoethanol, and 1 mM ATP. The samples were incubated at 25 °C for 30 min, and then analyzed by 4% non-denaturing PAGE in 0.5× TBE. The gel was stained with SYBR Green I solution, and the DNA was visualized by an iBright Imaging System (Thermo Fisher Scientific).

## ATPase assay

The 169 base-pair DNA fragments or 24N0 nucleosomes (0.06 μM) were mixed with DDM1 (0.48 μM) in a 50 μL total reaction volume, containing 54 mM Tris-HCl (pH 7.5), 15 mM NaCl, 0.1 mM DTT, 0.2 mM 2-mercaptoethanol, 1.5% glycerol, 0.5 mM ATP, and 2.5 mM MgCl$_2$, and incubated at 30 °C for 5, 15, 30, and 60 min. ATPase assays were performed using a colorimetric kit (abcam) and ATPase activity was calculated from optical density values at 600 nm using a Synergy H1M2F.

## Reporting summary

Further information on research design is available in the Nature Portfolio Reporting Summary linked to this article.

## Data availability

The cryo-EM maps and atomic models in this study have been deposited in the Electron Microscopy Data Bank and the Protein Data Bank, under the accession codes EMD- 36083 (DDM1-nucleosome complex) and PDB ID 8J90 (DDM1-nucleosome complex) for the AtDDM1-nucleosome complex, EMD-36084 (H2A nucleosome) and PDB ID 8J91 (H2A nucleosome) for the nucleosome containing AtH2A, and EMD-36085 (H2A.W nucleosome) and PDB ID 8J92 (H2A.W nucleosome) for the nucleosome containing AtH2A.W, respectively. The raw mass spectrometry data used in this study have been deposited to the proteomeXchange Consortium under accession code PXD043417 (Crosslinking mass spectrometry of DDM1 complexed with the nucleosome) via the Japan ProteOme STandard (JPOST) repository (JPST002218 (Crosslinking mass spectrometry of DDM1 complexed with the nucleosome))[118]. The structures of the nucleosome composed of the Widom 601 DNA sequence, human nucleosome core particle, Snf2-nucleosome complex, and DDM1-nucleosome complex used in this study can be found in the Protein Data Bank under the accession codes 3LZ0 (Nucleosome core particle composed of the Widom 601 DNA sequence), 7VZ4 (human nucleosome core particle), 5X0Y (Snf2-nucleosome complex), and 7UX9 (DDM1-nucleosome complex), respectively. Source data are provided with this paper.

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

## Acknowledgements

We thank all members of the Berger, Kakutani, and Kurumizaka laboratories, and especially Mitsuo Ogasawara (Univ. Tokyo) for technical assistance with cryo-EM data collection, and Y. Iikura, M. Dacher, and Y. Takeda (Univ. Tokyo) for their assistance. This work was supported in part by JSPS KAKENHI Grant Numbers JP21K20628, JP22H05172, and JP22H05178 [to A.O.], JP23H05475 [to H.K.], JP22K06098 [to Y.T.], and 21H04977 and 23H00365 [to T.K.], Research Support Project for Life Science and Drug Discovery (BINDS) from AMED under Grant Number JP23ama121009 [to H.K.], HFSP Grant Number RGP0025/2021 [to T.K.], JST ERATO Grant Number JPMJER1901 [to H.K.], and JST PRESTO Grant Number JPMJPR20K3 [to A.O.]. This work was also supported by the Austrian Science Fund (FWF): P32054 and P33380 [to F.B.].

## Author contributions

A.O., F.B., T.K., and H.K. conceived, designed, and supervised all the work. A.O. performed all the biochemical analyses. A.O., Y.T., and N.H. prepared the DDM1-nucleosome complex for cryo-EM. A.O. and Y.T. performed cryo-EM analyses. A.O., S.H., and L.N. performed crosslinking mass spectrometry. A.O. and S.S. performed the FRET assay. A.O., Y.T., F.B., T.K., and H.K. prepared all figures and wrote the manuscript. All of the authors discussed the results and commented on the manuscript.

## Competing interests

The authors declare no competing interests.
