## [Peer Review File · Nature Communications]

REVIEWER COMMENTS

Reviewer #1 (Remarks to the Author):

DDM1 is an Snf2-type chromatin remodeler that functions in DNA methylation maintenance and transposon silencing in Arabidopsis. DDM1 has been shown to bind and deposit histone variant H2A.W over transposons for silencing. Additionally, it has the ability to slide nucleosomes, which could provide access to other proteins in the context of heterochromatin. This manuscript presents a structural and biochemical characterization of DDM1 remodeling activity on H2A.W nucleosomes. Osakabe et al demonstrate that DDM1 preferentially slides H2A.W nucleosomes in an ATP-dependent manner. Structural characterization via CryoEM shows that DDM1 primarily binds nucleosomal DNA with minimal interaction with the nucleosome histone core (H4 tail interaction only). However, additional DDM1:nucleosome interactions were identified via crosslinking MS, including unique interactions with H2A.W tail. Mutational analysis demonstrates the importance of both H2W.A and H4 tail in nucleosome sliding assay. Overall, results show that DDM1 has a remodeling activity on H2A.W nucleosomes in addition to the previously known binding and deposition of this variant. While structural analysis is limited based on low resolution of the DDM1:nucleosome complex, the authors conducted insightful experiments via MS-XL and nucleosome sliding assays to support their conclusions. The data provide a strong basis for a remodeling activity of Arabidopsis DDM1. The manuscript is of high technical quality, but insight into DDM1 mechanism and function are somewhat limited. Comments and suggestions to further improve the manuscript are listed below.

Essential Revisions:

1. Figure 3a: Please state the local resolution range of DDM1 and nucleosome core in the text. Based on Supp Fig 2e, the density for DDM1 is 5-6+ Å. At this resolution, if you cannot confidently identify the position of amino acid side chains, they should not be shown in the figure. Additionally, this figure would be improved by including the density of the H4 tail, since histone tails are historically hard to identify due to their flexibility. As Snf2 is not in this figure, it is confusing to include the residue labels.
2. Please comment on the observation that DDM1 is primarily bound to the nucleosomal DNA and that there is no observed contact with histones, specifically H2A.W (i.e., state that the C-terminal tail is disordered in structure)

3. Figure 4d: WT nucleosome values are lower than previous panels (both H2A and H2A.W)?

a. H2A mutant “substantially enhanced” is ~12%, which is still less than 50% of H2A.W shown in previous panels at ~30% (still less than H2A.W in this panel too). What differences could account for this? this should be explained

b. Recommended to soften the statement that this is an “essential role” as activity is still observed at 10% without it.

c. Line 164 “DDM1 binds the H2A.W nucleosome through interactions with the specific H2A.W C-terminal residues” Binding assays with these mutants would strengthen this claim.

4. Supplemental Figure 5: Did MS verify the interactions with the H4 tail that were seen in the structure? Comment on why there are many observed crosslinks with histones and the structure only shows interaction with DNA?

5. An undiscussed topic that would add to the discussion is other histone variants and PTMs found in Arabidopsis. Jamge et al, 2023 found that H3 variants (H3.1/H3.3) form heterotypic nucleosomes and do not associate with a specific H2A variant. How do you anticipate this to effect DDM1 activity and the proposed model? Furthermore, they report that DDM1 uses the same conserved sites to bind both H2A.W and H2A.Z. Bourguet et al, 2022 found that H2A.W cooperates with H3 lysine 9 dimethylation. Expanding this topic in the discussion will place the new DDM1 mechanistic insight into the larger context of chromatin dynamics.

Additional Minor Comments:

1. For the nucleosome sliding assay, the author used terms such as “drastically higher” or “substantially enhanced.” These conclusions would be strengthened with a statistical analysis for significance.

2. Line 122 “The N-terminal tail of H4 is located near the ATPase core domain of DDM1.”

a. This statement would be strengthened by including distances in Figure 3a. As show, DDM1 residues are red residues and H4 is blue, this is misleading with the electrostatic potential scale in the same figure panel. What is charge of the H4 tail residues? Can you see more of this tail

compared to the nucleosome alone structure? Adding this discussion could strengthen claim that the H4 tail is bound in the DDM1 acidic pocket.

1. Figure 2b- what pdb is used for free nucleosome, specifically is it H2A or H2A.W?

2. Please include in the manuscript text if the entire DDM1 was used in the structure.

3. Figure 3b: What is the 80 bp band? Is there an explanation for its disappearance in the H2A.W nucleosome sample only?

4. Figure 3a: I think residue 557 should be a D, based on the sequence in Supp Fig 3.

5. Figure 4: What is the distance for each contact and the estimated length of the dashed line? Is it reasonable for the C-terminal tail to reach that far based on amino acid length? If possible, the dashed lines should be in the same position for each orientation and connect to the residue (yellow circle) (example: contact #3 in bottom right extends past the yellow circle)

6. Figure 4c: The authors note that the specific bands corresponding to DDM1:nucleosome complexes disappear in nucleosome lacking the N-term H4 tail, but there is still a clear shift? Why is there more of this band for H2A than H2A.W?

7. Line 145 “Two residues (K203 and K208) of DDM1 crosslinked to the C-terminal tail of H2A.W were close to the regions interacting with nucleotides in the complex Snf12/nucleosomes.”

a. Without more context, this sentence doesn't make sense. If you want to make a comparison to another structure, more explanation is required.

8. Fig 5: Where is the density of DDM1 in these figures? Indicate where DDM1 is sitting on DNA (arrow or asterisk if density isn't shown)

9. Figures 5d and S4d and e, could be a new supplemental figure to maintain the order in manuscript text

10. Please include in the figure legend that the nucleosome remodeling assay is normalized to - DDM1

Reviewer #2 (Remarks to the Author):

I co-reviewed this manuscript with one of the reviewers who provided the listed reports as part of the Nature Communications initiative to facilitate training in peer review and appropriate recognition for co-reviewers.

Reviewer #3 (Remarks to the Author):

Osakabe, Takizawa et al. report the structure of an H2A.W bound nucleosome and the structure of the *A. thaliana* chromatin remodeler DDM1 bound to the H2A.W nucleosome. DDM1 binds the nucleosomal substrate as observed for other chromatin remodelers. Additionally, the authors employ biochemical assays to show that DDM1 is able to shift nucleosomes and that nucleosome shifting by DDM1 is stimulated by the C-terminal tail of H2A.W. The presented work is mostly descriptive and contains limited amounts of additional mechanistic insight as the biochemical analysis remains very limited (cf. major comments). The cryo-EM data analysis is solid, although some important aspects to judge data quality are missing from the current manuscript (cf. minor comments).

Major comments

1. The present nucleosome shift assay is sufficient to demonstrate that DDM1 is able to shift nucleosomes and measure the effects of the C-terminal tail of the H2A variants/tailless H4. However, this assay is not state of the art to show nucleosome sliding as the read-out of the nucleosome shift is only indirect. The authors should repeat the assay and use an actual nucleosome sliding assay where the moved nucleosomes are directly observed through a shift on a NativePAGE gel. Otherwise, the authors cannot exclude that the observed generation of DNA truncations through the restriction digest is not due to secondary effects where the ATP simply induces a conformation shift in DDM1 that favors restriction digest.

2. The nucleosome shift assay is only giving very limited read-outs in regard to the mechanistic effect that the H2A.W tail has on DDM1 activity. The authors should additionally perform experiments that demonstrate if the H2A.W tail impacts ATPase hydrolysis rates or if the H2A.W tail only impacts the coupling of ATPase hydrolysis and DNA translocation. Same is true for the shift assay in the H2A and H2A.W comparison. Does the H2A.W nucleosome stimulate ATPase activity of DDM1 or is it simply easier to remodel H2A.W nucleosomes by achieving higher coupling rates? The authors should address these questions experimentally.

3. Much of the model that is presented in Fig. 6 is in no way explained by the data presented in the manuscript. If the authors include a model figure, they should focus solely on a model figure that directly pertains to their findings.

4. The authors show unwrapping of DNA in the presence of DDM1. It remains unclear if this is only a cryo-EM artifact or is indeed directly related to DDM1. Therefore, the authors should test via a biochemical/biophysical assay if binding of DDM1 to the nucleosome induces nucleosome unwrapping.

Minor comments

1. This reviewer disagrees with the adjective used in line 104 “drastically”. There is no doubt that H2A.W nucleosomes are more easily remodelled, but it seems to only be a modest effect (about 3X stimulation).

2. The authors should give precise numbers of how many rounds of 2D classification were performed (Classification trees).

3. The scale bar for 2D classes is missing for Supplementary Figure 2 and 7.

4. Map-To-Model FSC curves are missing.

5. 3D FSC plots are missing for both cryo-EM reconstructions.

6. Map-To-Densities figures are missing. These figures should show that key interactions (e.g., DNA distortion, H4 tail interaction etc.) are correctly modelled.

7. The authors should point out that there is competing work from the Martienssen lab (doi: 10.1101/2023.07.11.548598)

Reviewer #4 (Remarks to the Author):

In this manuscript, the authors study a chromatin remodeling factor Snf2 homolog in plant, named DDM1 (DECREASE IN DNA METHYLATION 1). DDM1 is known for deposit H2A.W variant in Arabidopsis. They determined cryo-EM structures of H2A.W nucleosome with and without bound DDM1, respectively at resolutions of 2.9Å and 4.7Å. The DDM1-bound nucleosome is significantly more disordered including only 111 base pairs of histone-bound DNA, instead of 145 base pairs. In the low-resolution DDM1-bound nucleosome structure, the authors observed interactions between H4 N-terminal basic residues and DDM1, but the interactions between the C-terminal H2A.W and DDM1 is absent, probably due to the disordered nature of the structure. They used crosslinking mass spectrometry to establish the DDM1-H2A.W interactions. Furthermore, the authors used mutagenesis of histones, N-terminal deletion mutant of H4 and swapping the C-terminal tails of H2A and H2A.W, and nucleosome sliding assays to conform the structural observations.

Overall, this is a well characterized study. One implication of the DDM1-induced flexibility is to increase the nucleosome accessibility of DNA binding proteins, including DNA methyltransferases. I will suggest that the authors perform DNA methylation assay using the nucleosome substrates in the presence and absence of DDM1.

In the last paragraph of Discussion, the authors suggested a similarity between Arabidopsis DDM1 and mammalian HELLS, in guiding DNA methylation. Is this similarity reflected in the amino acid sequence similarity between DDM1 and HELLS? If so, the sequence of HELLS should be included in the alignment of DDM1 and Snf2 shown in Figure S3. If not, the speculation should be made clear that DDM1 and HELLS do not share sequence similarity.

REVIEWER COMMENTS

Reviewer #1 (Remarks to the Author):

DDM1 is an Snf2-type chromatin remodeler that functions in DNA methylation maintenance and transposon silencing in Arabidopsis. DDM1 has been shown to bind and deposit histone variant H2A.W over transposons for silencing. Additionally, it has the ability to slide nucleosomes, which could provide access to other proteins in the context of heterochromatin. This manuscript presents a structural and biochemical characterization of DDM1 remodeling activity on H2A.W nucleosomes. Osakabe et al demonstrate that DDM1 preferentially slides H2A.W nucleosomes in an ATP-dependent manner. Structural characterization via CryoEM shows that DDM1 primarily binds nucleosomal DNA with minimal interaction with the nucleosome histone core (H4 tail interaction only). However, additional DDM1:nucleosome interactions were identified via crosslinking MS, including unique interactions with H2A.W tail. Mutational analysis demonstrates the importance of both H2W.A and H4 tail in nucleosome sliding assay. Overall, results show that DDM1 has a remodeling activity on H2A.W nucleosomes in addition to the previously known binding and deposition of this variant. While structural analysis is limited based on low resolution of the DDM1:nucleosome complex, the authors conducted insightful experiments via MS-XL and nucleosome sliding assays to support their conclusions. The data provide a strong basis for a remodeling activity of Arabidopsis DDM1. The manuscript is of high technical quality, but insight into DDM1 mechanism and function are somewhat limited. Comments and suggestions to further improve the manuscript are listed below.

Essential Revisions:

1. Figure 3a: Please state the local resolution range of DDM1 and nucleosome core in the text. Based on Supp Fig 2e, the density for DDM1 is 5-6+ Å. At this resolution, if you cannot confidently identify the position of amino acid side chains, they should not be shown in the figure. Additionally, this figure would be improved by including the density of the H4 tail, since histone tails are historically hard to identify due to their flexibility. As Snf2 is not in this figure, it is confusing to include the residue labels.
2. Please comment on the observation that DDM1 is primarily bound to the nucleosomal DNA and that there is no observed contact with histones, specifically H2A.W (i.e., state that the C-terminal tail is disordered in structure)
3. Figure 4d: WT nucleosome values are lower than previous panels (both H2A and H2A.W)?

a. H2A mutant “substantially enhanced” is ~12%, which is still less than 50% of H2A.W shown in previous panels at ~30% (still less than H2A.W in this panel too). What differences could account for this? this should be explained

b. Recommended to soften the statement that this is an “essential role” as activity is still observed at 10% without it.

c. Line 164 “DDM1 binds the H2A.W nucleosome through interactions with the specific H2A.W C-terminal residues” Binding assays with these mutants would strengthen this claim.

4. Supplemental Figure 5: Did MS verify the interactions with the H4 tail that were seen in the structure? Comment on why there are many observed crosslinks with histones and the structure only shows interaction with DNA?

5. An undiscussed topic that would add to the discussion is other histone variants and PTMs found in Arabidopsis. Jamge et al, 2023 found that H3 variants (H3.1/H3.3) form heterotypic nucleosomes and do not associate with a specific H2A variant. How do you anticipate this to effect DDM1 activity and the proposed model? Furthermore, they report that DDM1 uses the same conserved sites to bind both H2A.W and H2A.Z. Bourguet et al, 2022 found that H2A.W cooperates with H3 lysine 9 dimethylation. Expanding this topic in the discussion will place the new DDM1 mechanistic insight into the larger context of chromatin dynamics.

Additional Minor Comments:

1. For the nucleosome sliding assay, the author used terms such as “drastically higher” or “substantially enhanced.” These conclusions would be strengthened with a statistical analysis for significance.

2. Line 122 “The N-terminal tail of H4 is located near the ATPase core domain of DDM1.”

a. This statement would be strengthened by including distances in Figure 3a. As show, DDM1 residues are red residues and H4 is blue, this is misleading with the electrostatic potential scale in the same figure panel. What is charge of the H4 tail residues? Can you see more of this tail compared to the nucleosome alone structure? Adding this discussion could strengthen claim that the H4 tail is bound in the DDM1 acidic pocket.

1. Figure 2b- what pdb is used for free nucleosome, specifically is it H2A or H2A.W?

2. Please include in the manuscript text if the entire DDM1 was used in the structure.

3. Figure 3b: What is the 80 bp band? Is there an explanation for its disappearance in the H2A.W nucleosome sample only?

4. Figure 3a: I think residue 557 should be a D, based on the sequence in Supp Fig 3.

5. Figure 4: What is the distance for each contact and the estimated length of the dashed line? Is it reasonable for the C-terminal tail to reach that far based on amino acid length? If possible, the dashed lines should be in the same position for each orientation and connect to the residue (yellow circle) (example: contact #3 in bottom right extends past the yellow circle)

6. Figure 4c: The authors note that the specific bands corresponding to DDM1:nucleosome complexes disappear in nucleosome lacking the N-term H4 tail, but there is still a clear shift? Why is there more of this band for H2A than H2A.W?

7. Line 145 “Two residues (K203 and K208) of DDM1 crosslinked to the C-terminal tail of H2A.W were close to the regions interacting with nucleotides in the complex Snf12/nucleosomes.”
 - a. Without more context, this sentence doesn't make sense. If you want to make a comparison to another structure, more explanation is required.

8. Fig 5: Where is the density of DDM1 in these figures? Indicate where DDM1 is sitting on DNA (arrow or asterisk if density isn't shown)

9. Figures 5d and S4d and e, could be a new supplemental figure to maintain the order in manuscript text

10. Please include in the figure legend that the nucleosome remodeling assay is normalized to -DDM1

Reviewer #2 (Remarks to the Author):

I co-reviewed this manuscript with one of the reviewers who provided the listed reports as part of the Nature Communications initiative to facilitate training in peer review and appropriate recognition for co-

reviewers.

Reviewer #3 (Remarks to the Author):

Osakabe, Takizawa et al. report the structure of an H2A.W bound nucleosome and the structure of the *A. thaliana* chromatin remodeler DDM1 bound to the H2A.W nucleosome. DDM1 binds the nucleosomal substrate as observed for other chromatin remodelers. Additionally, the authors employ biochemical assays to show that DDM1 is able to shift nucleosomes and that nucleosome shifting by DDM1 is stimulated by the C-terminal tail of H2A.W. The presented work is mostly descriptive and contains limited amounts of additional mechanistic insight as the biochemical analysis remains very limited (cf. major comments). The cryo-EM data analysis is solid, although some important aspects to judge data quality are missing from the current manuscript (cf. minor comments).

Major comments

1. The present nucleosome shift assay is sufficient to demonstrate that DDM1 is able to shift nucleosomes and measure the effects of the C-terminal tail of the H2A variants/tailless H4. However, this assay is not state of the art to show nucleosome sliding as the read-out of the nucleosome shift is only indirect. The authors should repeat the assay and use an actual nucleosome sliding assay where the moved nucleosomes are directly observed through a shift on a NativePAGE gel. Otherwise, the authors cannot exclude that the observed generation of DNA truncations through the restriction digest is not due to secondary effects where the ATP simply induces a conformation shift in DDM1 that favors restriction digest.

2. The nucleosome shift assay is only giving very limited read-outs in regard to the mechanistic effect that the H2A.W tail has on DDM1 activity. The authors should additionally perform experiments that demonstrate if the H2A.W tail impacts ATPase hydrolysis rates or if the H2A.W tail only impacts the coupling of ATPase hydrolysis and DNA translocation. Same is true for the shift assay in the H2A and H2A.W comparison. Does the H2A.W nucleosome stimulate ATPase activity of DDM1 or is it simply easier to remodel H2A.W nucleosomes by achieving higher coupling rates? The authors should address these questions experimentally.

3. Much of the model that is presented in Fig. 6 is in no way explained by the data presented in the manuscript. If the authors include a model figure, they should focus solely on a model figure that directly pertains to their findings.

4. The authors show unwrapping of DNA in the presence of DDM1. It remains unclear if this is only a cryo-EM artifact or is indeed directly related to DDM1. Therefore, the authors should test via a biochemical/biophysical assay if binding of DDM1 to the nucleosome induces nucleosome unwrapping.

Minor comments

1. This reviewer disagrees with the adjective used in line 104 “drastically”. There is no doubt that H2A.W nucleosomes are more easily remodelled, but it seems to only be a modest effect (about 3X stimulation).

2. The authors should give precise numbers of how many rounds of 2D classification were performed (Classification trees).

3. The scale bar for 2D classes is missing for Supplementary Figure 2 and 7.

4. Map-To-Model FSC curves are missing.

5. 3D FSC plots are missing for both cryo-EM reconstructions.

6. Map-To-Densities figures are missing. These figures should show that key interactions (e.g., DNA distortion, H4 tail interaction etc.) are correctly modelled.

7. The authors should point out that there is competing work from the Martienssen lab (doi: 10.1101/2023.07.11.548598)

Reviewer #4 (Remarks to the Author):

In this manuscript, the authors study a chromatin remodeling factor Snf2 homolog in plant, named DDM1 (DECREASE IN DNA METHYLATION 1). DDM1 is known for deposit H2A.W variant in Arabidopsis. They determined cryo-EM structures of H2A.W nucleosome with and without bound DDM1, respectively at resolutions of 2.9Å and 4.7Å. The DDM1-bound nucleosome is significantly more disordered including only 111 base pairs of histone-bound DNA, instead of 145 base pairs. In the low-resolution DDM1-bound nucleosome structure, the authors observed interactions between H4 N-terminal basic residues and DDM1, but the interactions between the C-terminal H2A.W and DDM1 is absent, probably due to the

disordered nature of the structure. They used crosslinking mass spectrometry to establish the DDM1-H2A.W interactions. Furthermore, the authors used mutagenesis of histones, N-terminal deletion mutant of H4 and swapping the C-terminal tails of H2A and H2A.W, and nucleosome sliding assays to conform the structural observations.

Overall, this is a well characterized study. One implication of the DDM1-induced flexibility is to increase the nucleosome accessibility of DNA binding proteins, including DNA methyltransferases. I will suggest that the authors perform DNA methylation assay using the nucleosome substrates in the presence and absence of DDM1.

In the last paragraph of Discussion, the authors suggested a similarity between Arabidopsis DDM1 and mammalian HELLS, in guiding DNA methylation. Is this similarity reflected in the amino acid sequence similarity between DDM1 and HELLS? If so, the sequence of HELLS should be included in the alignment of DDM1 and Snf2 shown in Figure S3. If not, the speculation should be made clear that DDM1 and HELLS do not share sequence similarity.

REVIEWER COMMENTS

Reviewer #1 (Remarks to the Author):

DDM1 is an Snf2-type chromatin remodeler that functions in DNA methylation maintenance and transposon silencing in Arabidopsis. DDM1 has been shown to bind and deposit histone variant H2A.W over transposons for silencing. Additionally, it has the ability to slide nucleosomes, which could provide access to other proteins in the context of heterochromatin. This manuscript presents a structural and biochemical characterization of DDM1 remodeling activity on H2A.W nucleosomes. Osakabe et al demonstrate that DDM1 preferentially slides H2A.W nucleosomes in an ATP-dependent manner. Structural characterization via CryoEM shows that DDM1 primarily binds nucleosomal DNA with minimal interaction with the nucleosome histone core (H4 tail interaction only). However, additional DDM1:nucleosome interactions were identified via crosslinking MS, including unique interactions with H2A.W tail. Mutational analysis demonstrates the importance of both H2W.A and H4 tail in nucleosome sliding assay. Overall, results show that DDM1 has a remodeling activity on H2A.W nucleosomes in addition to the previously known binding and deposition of this variant. While structural analysis is limited based on low resolution of the DDM1:nucleosome complex, the authors conducted insightful experiments via MS-XL and nucleosome sliding assays to support their conclusions. The data provide a strong basis for a remodeling activity of Arabidopsis DDM1. The manuscript is of high technical quality, but insight into DDM1 mechanism and function are somewhat limited. Comments and suggestions to further improve the manuscript are listed below.

Essential Revisions:

Comment 1)

1. Figure 3a: Please state the local resolution range of DDM1 and nucleosome core in the text. Based on Supp Fig 2e, the density for DDM1 is 5-6+ Å. At this resolution, if you cannot confidently identify the position of amino acid side chains, they should not be shown in the figure. Additionally, this figure would be improved by including the density of the H4 tail, since histone tails are historically hard to identify due to their flexibility. As Snf2 is not in this figure, it is confusing to include the residue labels.

Reply)

We thank the reviewer for this comment. In the revised manuscript, we described the local resolution of DDM1 and the nucleosome core in the text (p.6, ll.20-24). As

Reviewer #1 suggested, we removed the position of the amino acid side chain due to the low resolution of the DDM1-H2A.W nucleosome complex structure. We also showed the H4 tail densities in the H2A.W nucleosome and DDM1-H2A.W nucleosome complex for the comparison of the H4 tail structures, and removed the residue labels of Snf2 to avoid confusion. These figures are now shown in Fig. 7c and Supplementary Fig. 13 of the revised manuscript.

Comment 2)

2. Please comment on the observation that DDM1 is primarily bound to the nucleosomal DNA and that there is no observed contact with histones, specifically H2A.W (i.e., state that the C-terminal tail is disordered in structure)

Reply)

We appreciate Reviewer #1's request to clarify our observations in the cryo-EM structure of the DDM1-H2A.W nucleosome complex. We described the disordered structure of the C-terminal tail of H2A.W and the lack of interactions between DDM1 and H2A.W in the cryo-EM structure of the DDM1-H2A.W nucleosome complex in the text (p.9, ll.2-5).

Comment 3)

3. Figure 4d: WT nucleosome values are lower than previous panels (both H2A and H2A.W)?

Reply)

In the process of the revision that Reviewer #3 suggested, we performed the assay to directly detect the nucleosome sliding on the DNA. We then found that DDM1 slides on nucleosomes containing H2A and H2A.W with the same efficiency. The previous nucleosome sliding assay with restriction enzymes may have detected the lower flexibility of the entry/exit nucleosomal DNA specifically occurring in the H2A.W nucleosome, but not the H2A nucleosome. Therefore, we removed all results regarding the nucleosome sliding assay with restriction enzymes in the revised manuscript. The new results of the "REAL nucleosome sliding assay" are presented in Fig. 7 of the revised manuscript.

Comment 3a)

a. H2A mutant "substantially enhanced" is ~12%, which is still less than 50% of

H2A.W shown in previous panels at ~30% (still less than H2A.W in this panel too).
What differences could account for this? this should be explained

Reply)

Again, as explained above, we removed all descriptions regarding the results of nucleosome sliding assay with the restriction enzyme.

Comment 3b)

b. Recommended to soften the statement that this is an “essential role” as activity is still observed at 10% without it.

Reply)

We removed this statement accordingly.

Comment 3c)

c. Line 164 “DDM1 binds the H2A.W nucleosome through interactions with the specific H2A.W C-terminal residues” Binding assays with these mutants would strengthen this claim.

Reply)

We removed the results of the nucleosome sliding assay with mutant nucleosomes containing C-terminal tail-swapped H2A variants in the revised manuscript. However, we observed the potential interaction between DDM1 and the H2A.W C-terminal tail by crosslinking mass spectrometry, as shown in Fig. 6 and Supplementary Fig. 10 of the revised manuscript. We then confirmed that these interactions are structurally possible in the DDM1-H2A.W nucleosome complex. We stated this point in the text (p.9, ll.8-22).

Comment 4)

4. Supplemental Figure 5: Did MS verify the interactions with the H4 tail that were seen in the structure? Comment on why there are many observed crosslinks with histones and the structure only shows interaction with DNA?

Reply)

We thank the reviewer for this comment. Indeed, our crosslinking mass spectrometric analyses did not detect the interaction between the N-terminal tail of H4 and the acidic

pocket of DDM1, in contrast to our observations of the cryo-EM structure of the DDM1-H2A.W nucleosome complex. We reason that the acidic pocket of DDM1 is enriched with acidic residues and devoid of lysine residues, and therefore we could not detect the crosslinking by DSS-H12/D12. This is mentioned in the text (p.12, ll.18-20). Most of the crosslinking interactions are observed between the disordered tail regions of the histone and DDM1. The disordered histone tails are not visible in the cryo-EM technique. This is the reason why many observed crosslinks (most of invisible histone tails) cannot be visualized by cryo-EM. It should be noted that the crosslinking interactions in the substructural classes, which are discarded during the three-dimensional reconstruction, may also be detected in our crosslinking experiments. In the revised manuscript, we described the possible interaction area of the nucleosomal H2A.W C-terminal tail, and revealed that the entire DDM1 region can be interact with it. This result is presented in the new Fig. 6b. We removed the crosslinking mass spectrometry results for the DDM1-H2A nucleosome because we did not obtain the cryo-EM structure of the DDM1-H2A nucleosome complex. Finally, we detected two possible crosslinks between DDM1 and H2A.W, as shown in Fig. 6, and one crosslink between DDM1 and H2B, as shown in Supplementary Fig. 10.

Comment 5)

5. An undiscussed topic that would add to the discussion is other histone variants and PTMs found in Arabidopsis. Jamge et al, 2023 found that H3 variants (H3.1/H3.3) form heterotypic nucleosomes and do not associate with a specific H2A variant. How do you anticipate this to effect DDM1 activity and the proposed model? Furthermore, they report that DDM1 uses the same conserved sites to bind both H2A.W and H2A.Z. Bourguet et al, 2022 found that H2A.W cooperates with H3 lysine 9 dimethylation. Expanding this topic in the discussion will place the new DDM1 mechanistic insight into the larger context of chromatin dynamics.

Reply)

This is an important issue and we thank Reviewer #1 for raising it. Our cryo-EM structure of the DDM1-H2A.W nucleosome complex and the current nucleosome sliding assay results indicate that DDM1 requires only the N-terminal tail of H4, and does not require H2A variants to slide nucleosomes. As a previous study indicated that H4 tail acetylation weakens the interaction with DDM1 (Lee *et al.*, 2023, *Cell*), at least the PTMs of H4 contribute to the nucleosome sliding activity of DDM1. This result was supported by our current findings, showing that the nucleosome sliding activity of

DDM1 was decreased by the removal of the N-terminal tail of H4, as shown in Fig. 7d and e in the revised manuscript. In our previous biochemical study, the same regions of DDM1 bound both H2A.W and H2A.Z in a pull-down assay using histone dimers, but not nucleosomes. In contrast, we did not observe the interaction between DDM1 and H2A variants within nucleosomes in our cryo-EM structure of the DDM1-H2A.W nucleosome complex. These results suggest that we could separate the function of DDM1 into two roles: 1) the deposition of H2A.W onto transposons, and 2) the sliding of nucleosomes regardless of H2A variants. In addition, our crosslinking mass spectrometric analyses identified the interaction between the N-terminal tail of H3 and the C-terminal tail of H2A.W. This observation may be important to understand the link between H2A.W and H3 lysine 9 dimethylation for transposon silencing, which would contribute to a thorough discussion regarding the mechanistic insights about the establishment or maintenance of repressive epigenetic marks over transposons. We discussed our observation of the interaction between the N-terminal tail of H3 and the C-terminal tail of H2A.W and its possible role in transposon silencing in the revised manuscript (p.12, l.21- p.13, l.4).

Additional Minor Comments:

1. For the nucleosome sliding assay, the author used terms such as “drastically higher” or “substantially enhanced.” These conclusions would be strengthened with a statistical analysis for significance.

Reply)

As we removed the results of nucleosome sliding assay with restriction enzyme, we did not use such phrases pointed by Reviewer #1 in the revised manuscript.

2. Line 122 “The N-terminal tail of H4 is located near the ATPase core domain of DDM1.”

a. This statement would be strengthened by including distances in Figure 3a. As show, DDM1 residues are red residues and H4 is blue, this is misleading with the electrostatic potential scale in the same figure panel. What is charge of the H4 tail residues? Can you see more of this tail compared to the nucleosome alone structure? Adding this discussion could strengthen claim that the H4 tail is bound in the DDM1 acidic pocket.

Reply)

We thank Reviewer #1 for this constructive comment. First, we measured and showed the distance of the C α atoms between the residue in the H4 tail and its possible binding residue in DDM1, because the resolution of our DDM1-nucleosome cryo-EM structure is 4.7 Å, which is insufficient for identifying the side chains. These new data are shown in Supplementary Fig. 13b. Second, we changed the colors of H4 and DDM1 to avoid confusion with the electrostatic potential scale, and showed the electrostatic potential scales of H4 tail and DDM1 in a different panel, Fig. 7c. Third, we presented the structure of the DDM1-free nucleosome next to the DDM1-nucleosome complex to emphasize our observation that the N-terminal tail of H4 in the nucleosome alone was disordered, but detected by an interaction with DDM1's acidic pocket. These new data are also shown in Fig. 7c.

1. Figure 2b- what pdb is used for free nucleosome, specifically is it H2A or H2A.W?

Reply)

We used the H2A.W nucleosome for the comparison of the nucleosomal DNA structure around SHL-2. We mentioned the use of the H2A.W nucleosome in Fig. 2b and its legend.

2. Please include in the manuscript text if the entire DDM1 was used in the structure.

Reply)

As Reviewer #1 suggested, we mentioned that the entire/full length DDM1 was used for the cryo-EM structure in the text (p.6, ll.16-19).

3. Figure 3b: What is the 80 bp band? Is there an explanation for its disappearance in the H2A.W nucleosome sample only?

Reply)

The 80 bp band indicates the product after cleavage by Bsh1236I, which recognizes the sequence close to the dyad axis of the nucleosomal DNA, meaning that this enzyme cleaved nucleosome-free DNA. However, as we removed the nucleosome sliding assay with restriction enzymes, the previous Figure 3b is not shown in the revised manuscript.

4. Figure 3a: I think residue 557 should be a D, based on the sequence in Supp Fig 3.

Reply)

Since the Asp 557 residue is located far away from the Arg19 residue of the H4 N-terminal tail detected in our cryo-EM structure, we removed our discussion about this residue in the revised manuscript, as shown in Supplementary Fig. 13b.

5. Figure 4: What is the distance for each contact and the estimated length of the dashed line? Is it reasonable for the C-terminal tail to reach that far based on amino acid length? If possible, the dashed lines should be in the same position for each orientation and connect to the residue (yellow circle) (example: contact #3 in bottom right extends past the yellow circle)

Reply)

We thank Reviewer #1 for this critical comment. We carefully checked the possible interaction between DDM1 and the C-terminal tail of H2A.W by making the 115.35 Å radius corresponding to residues 113-140 of H2A.W (the central point is the C α atom of His113 of H2A.W), which indicates the possible crosslinking area of the H2A.W Lys140 by DSS-H12/D12. Our new approach revealed that the Lys140 and Lys147 residues of H2A.W could contact the Lys208 and Lys342 residues of DDM1, respectively. These new results are shown in Fig. 6 of the revised manuscript.

6. Figure 4c: The authors note that the specific bands corresponding to DDM1:nucleosome complexes disappear in nucleosome lacking the N-term H4 tail, but there is still a clear shift? Why is there more of this band for H2A than H2A.W?

Reply)

We supposed that the observation of multiple bands of the DDM1-nucleosome complex might reflect the various binding modes of DDM1 to nucleosomes as shown in Supplementary Fig. 6c, in which we detected extra densities in addition to the nucleosome by 3D classification. We prepared new nucleosomes with and without the N-terminal tails of the H4 nucleosome and performed the electrophoresis mobility shift assay. Our current results indicated no clear binding differences between H2A.W nucleosomes with or without this tail, as shown in Supplementary Fig. 14. Therefore, we removed the description about the disappearance of the band corresponding to the DDM1-nucleosome complex to avoid confusion.

7. Line 145 “Two residues (K203 and K208) of DDM1 crosslinked to the C-terminal

tail of H2A.W were close to the regions interacting with nucleotides in the complex Snf12/nucleosomes.”

a. Without more context, this sentence doesn't make sense. If you want to make a comparison to another structure, more explanation is required.

Reply)

In the first version of the submitted manuscript, we intended to present that the C-terminal tail of H2A.W might upregulate the ATPase activity of DDM1, such that DDM1 specifically slides the H2A.W nucleosome. However, as we have now described throughout our responses, our current biochemical results suggested that the ATPase activity of DDM1 with H2A.W was similar to that with H2A, and DDM1 slides nucleosomes containing both H2A and H2A.W with almost the same efficiency, as shown in Fig. 7 and Supplementary Fig. 12. Therefore, we removed this description to avoid confusion.

8. Fig 5: Where is the density of DDM1 in these figures? Indicate where DDM1 is sitting on DNA (arrow or asterisk if density isn't shown)

Reply)

In Fig. 3 of the revised manuscript, we show the cryo-EM density of DDM1 in the DDM1-nucleosome complex to clarify where DDM1 binds nucleosomes, for the structural comparison of the entry/exit nucleosomal DNA ends between DDM1-bound and DDM1-free nucleosomes.

9. Figures 5d and S4d and e, could be a new supplemental figure to maintain the order in manuscript text

Reply)

We moved these figures to the supplementary figures accordingly. In the revised manuscript, the previous Fig. 5d and Supplementary Fig. 4d and 4e are now shown as Supplementary Figs. 16, 15a, and 15b, respectively.

10. Please include in the figure legend that the nucleosome remodeling assay is normalized to -DDM1

Reply)

We included the description about the normalization of the nucleosome remodeling assay in the figure legend of Fig. 7.

Reviewer #2 (Remarks to the Author):

I co-reviewed this manuscript with one of the reviewers who provided the listed reports as part of the Nature Communications initiative to facilitate training in peer review and appropriate recognition for co-reviewers.

We thank Reviewer #2 for the contribution to peer review of our manuscript. We hope that we addressed all of your concerns in the revised manuscript.

Reviewer #3 (Remarks to the Author):

Osakabe, Takizawa et al. report the structure of an H2A.W bound nucleosome and the structure of the *A. thaliana* chromatin remodeler DDM1 bound to the H2A.W nucleosome. DDM1 binds the nucleosomal substrate as observed for other chromatin remodelers. Additionally, the authors employ biochemical assays to show that DDM1 is able to shift nucleosomes and that nucleosome shifting by DDM1 is stimulated by the C-terminal tail of H2A.W. The presented work is mostly descriptive and contains limited amounts of additional mechanistic insight as the biochemical analysis remains very limited (cf. major comments). The cryo-EM data analysis is solid, although some important aspects to judge data quality are missing from the current manuscript (cf. minor comments).

Major comments

Comment 1)

1. The present nucleosome shift assay is sufficient to demonstrate that DDM1 is able to shift nucleosomes and measure the effects of the C-terminal tail of the H2A variants/tailless H4. However, this assay is not state of the art to show nucleosome sliding as the read-out of the nucleosome shift is only indirect. The authors should repeat the assay and use an actual nucleosome sliding assay where the moved nucleosomes are directly observed through a shift on a NativePAGE gel. Otherwise, the authors cannot exclude that the observed generation of DNA truncations through the restriction digest is not due to secondary effects where the ATP simply induces a conformation shift in DDM1 that favors restriction digest.

Reply)

We really appreciate this comment of Reviewer #3 for revision. We set up and performed the new experiments to monitor the nucleosome sliding activity of DDM1 without a restriction enzyme. As a result, we found that our previous observation detected the lower flexibility of the entry/exit nucleosomal DNA ends of H2A.W, but we interpreted this result as the H2A.W-specific nucleosome sliding activity. This inconsistent result might be because our previous results reflected the low flexibility of the entry/exit nucleosomal DNA ends of the H2A.W nucleosome compared to the H2A nucleosome. In the revised manuscript, we removed all results of the nucleosome remodeling assay with restriction enzymes. Instead, we performed the actual nucleosome sliding assay as this reviewer suggested, and found that DDM1 slides nucleosomes containing both H2A and H2A.W with the same efficiency. These new results are now shown as Fig. 7 in the revised manuscript.

Comment 2)

2. The nucleosome shift assay is only giving very limited read-outs in regard to the mechanistic effect that the H2A.W tail has on DDM1 activity. The authors should additionally perform experiments that demonstrate if the H2A.W tail impacts ATPase hydrolysis rates or if the H2A.W tail only impacts the coupling of ATPase hydrolysis and DNA translocation. Same is true for the shift assay in the H2A and H2A.W comparison. Does the H2A.W nucleosome stimulate ATPase activity of DDM1 or is it simply easier to remodel H2A.W nucleosomes by achieving higher coupling rates? The authors should address these questions experimentally.

Reply)

We thank Reviewer #3 for this critical comment. We performed the ATPase assay with nucleosomes containing H2A and H2A.W. Consistent with our previous observation (Osakabe *et al.*, 2021, *Nat. Cell Biol.*), DDM1 showed the ATPase activity with DNA compared to DDM1 alone. In addition, we observed higher ATPase activity with nucleosomes rather than DNA, and this activity was detected with both the H2A and H2A.W nucleosomes, as shown in Supplementary Fig. 12 of the revised manuscript. These results also support our current observation of the same nucleosome sliding efficiency of DDM1 for the nucleosomes containing H2A and H2A.W, as shown in Fig. 7a and b of the revised manuscript.

Comment 3)

3. Much of the model that is presented in Fig. 6 is in no way explained by the data

presented in the manuscript. If the authors include a model figure, they should focus solely on a model figure that directly pertains to their findings.

Reply)

We changed the model based on our results, as shown in Fig. 8 of the revised manuscript.

4. The authors show unwrapping of DNA in the presence of DDM1. It remains unclear if this is only a cryo-EM artifact or is indeed directly related to DDM1. Therefore, the authors should test via a biochemical/biophysical assay if binding of DDM1 to the nucleosome induces nucleosome unwrapping.

Reply)

We thank the reviewer for this critical comment. We performed the restriction enzyme susceptibility assay and FRET assay to investigate if the flexibility of the entry/exit nucleosomal DNA ends is increased by DDM1 in solution. Our restriction enzyme susceptibility assay results suggested that DDM1 increased the DNA unwrapping from the histone octamer at the entry/exit regions of the nucleosome without disassembly of nucleosome, as shown in Fig. 4 of the revised manuscript. Furthermore, our FRET assay showed the same trends as our observations with the restriction enzyme susceptibility assay, as shown in Fig. 5.

Minor comments

1. This reviewer disagrees with the adjective used in line 104 “drastically”. There is no doubt that H2A.W nucleosomes are more easily remodelled, but it seems to only be a modest effect (about 3X stimulation).

Reply)

Since we did not observe the H2A.W-specific nucleosome sliding activity with the new experiments, as shown in Fig. 7, we removed this description.

2. The authors should give precise numbers of how many rounds of 2D classification were performed (Classification trees).

Reply)

We added the numbers of 2D classification rounds, as shown in Supplementary Figs. 2, 3, and 6 of the revised manuscript.

3. The scale bar for 2D classes is missing for Supplementary Figure 2 and 7.

Reply)

We added the scale bar for the 2D classes shown in Supplementary Figs. 2, 3, and 6 of the revised manuscript.

4. Map-To-Model FSC curves are missing.

Reply)

We added the Map-To-Model FSC curves, as shown in Supplementary Figs. 2, 3, and 6 of the revised manuscript.

5. 3D FSC plots are missing for both cryo-EM reconstructions.

Reply)

We added the 3D FSC plots, as shown in Supplementary Figs. 2, 3, and 6 of the revised manuscript.

6. Map-To-Densities figures are missing. These figures should show that key interactions (e.g., DNA distortion, H4 tail interaction etc.) are correctly modelled.

Reply)

We showed the Map-To-Densities figures for the interaction between H2A.W and H3, the DNA distortion, and the interaction between the H4 tail and DDM1, as shown in Supplementary Figs. 4, 7, and 13 of the revised manuscript, respectively.

7. The authors should point out that there is competing work from the Martienssen lab (doi: 10.1101/2023.07.11.548598)

Reply)

We mentioned and described the structural differences between the work from the Martienssen lab, the Du lab, and our current study in the text with Supplementary Fig.

17. Since the PDB file published by the Du lab is not available yet, we used the structure of the DDM1-nucleosome complex (PDB ID: 7UX9) for the comparison. We demonstrated that their observations for the binding mode of DDM1 to nucleosomes were almost the same as our results, while we found different structures for the loop of DDM1 facing toward H3. Intriguingly, the flexible features of the entry/exit nucleosomal DNA ends were observed only in our cryo-EM structure. We reason that this discrepancy might come from the use of histone H4 from different species (Martienssen lab used *Xenopus* H4) for the nucleosome reconstitution. It should be noted that there are two amino acid differences between *Xenopus* H4 and *Arabidopsis* H4, and these substitutions are located in the structurally important central helix and the DNA-binding loop. These facts are discussed in the revised manuscript (p.13, 1.16- p.14, 1.6).

Reviewer #4 (Remarks to the Author):

In this manuscript, the authors study a chromatin remodeling factor Snf2 homolog in plant, named DDM1 (DECREASE IN DNA METHYLATION 1). DDM1 is known for deposit H2A.W variant in *Arabidopsis*. They determined cryo-EM structures of H2A.W nucleosome with and without bound DDM1, respectively at resolutions of 2.9Å and 4.7Å. The DDM1-bound nucleosome is significantly more disordered including only 111 base pairs of histone-bound DNA, instead of 145 base pairs. In the low-resolution DDM1-bound nucleosome structure, the authors observed interactions between H4 N-terminal basic residues and DDM1, but the interactions between the C-terminal H2A.W and DDM1 is absent, probably due to the disordered nature of the structure. They used crosslinking mass spectrometry to establish the DDM1-H2A.W interactions. Furthermore, the authors used mutagenesis of histones, N-terminal deletion mutant of H4 and swapping the C-terminal tails of H2A and H2A.W, and nucleosome sliding assays to conform the structural observations.

Comment 1)

Overall, this is a well characterized study. One implication of the DDM1-induced flexibility is to increase the nucleosome accessibility of DNA binding proteins, including DNA methyltransferases. I will suggest that the authors perform DNA methylation assay using the nucleosome substrates in the presence and absence of DDM1.

Reply)

We thank Reviewer #4 for their positive comments. We completely agree with the point that the DNA methylation assay using a reconstituted nucleosome would clarify the mechanism by which the nucleosome sliding and unwrapping activity of DDM1 affect the maintenance of DNA methylation. However, it is still difficult to address this point because it requires a nucleosome containing hemi-methylated DNA and some factors including DNA methyltransferases such as MET1, for which preparation methods have not been established yet. Therefore, this point is very important but should be solved as a future issue.

In the last paragraph of Discussion, the authors suggested a similarity between Arabidopsis DDM1 and mammalian HELLS, in guiding DNA methylation. Is this similarity reflected in the amino acid sequence similarity between DDM1 and HELLS? If so, the sequence of HELLS should be included in the alignment of DDM1 and Snf2 shown in Figure S3. If not, the speculation should be made clear that DDM1 and HELLS do not share sequence similarity.

Reply)

We thank Reviewer #4 for this constructive comment. In our previous study (Osakabe *et al.*, 2021, *Nat. Cell Biol.*), we showed that the histone binding regions of DDM1 share conserved regions with mammalian HELLS/LSH. In this study, we identified possible regulatory regions in DDM1, like the AutoN domain of ISWI. Surprisingly, mammalian HELLS/LSH also showed some sequence similarities with these regulatory regions. We mentioned this point in the discussion of the text with Supplementary Fig. 15 (p.14, ll.7-18).

REVIEWERS' COMMENTS

Reviewer #1 (Remarks to the Author):

Overall, we feel that this manuscript was significantly improved in response to the review process. The new experiments and analyses changed the main conclusions, which highlights that they were necessary before publication. Since the initial submission, the publication of competing work has somewhat limited the novelty of this work. However, we feel that our comments have been adequately addressed and the manuscript can be accepted for publication.

Main Conclusions of Original Manuscript:

1. DDM1 preferentially slides H2A.W
 - o In revision: New sliding assay, now reporting DMM1 slides H2A and H2A.W with the same efficiency
2. H2A.W C-terminal tails and H4 N-terminal tail play important roles in DDM1-mediated nucleosome sliding
 - o In revision: Removed tail swap experiments completely, only show H4-tailless H2A.W
3. DDM1 binds nucleosomal DNA at SHL-2 and SHL+6
 - o In revision: Same conclusions for the DDM1-H2A.W structure
4. DDM1 contacts the H2A.W nucleosome and renders the entry/exit DNA regions flexibly disordered in the H2A.W nucleosome
 - o In revision: Better description of XL-MS data and more analysis of nucleosome flexibility

Main Conclusions of Revision:

1. H2A flexible entry/exit DNA regions compared to H2A.W (not really earth-shattering)
2. DDM1-H2A.W contacts H4 tail and DNA, increases entry/exit DNA flexibility to resemble H2A nucleosome
3. DDMI binds and slides H2A and H2A.W nucleosomes with the same efficiency

Reviewer #2 (Remarks to the Author):

Reviewer #3 (Remarks to the Author):

The authors have addressed by concerns sufficiently, and this reviewer is pleased that the addition of nucleosome sliding assays and ATPase assays could clarify how DDM1 works on nucleosomal substrates.

Reviewer #4 (Remarks to the Author):

The authors addressed partially my concerns. The DNA methylation assay is not feasible at this moment. I support the publication.

** See Nature Portfolio's author and referees' website at www.nature.com/authors for information about policies, services and author benefits

REVIEWER COMMENTS

Reviewer #1 (Remarks to the Author):

DDM1 is an Snf2-type chromatin remodeler that functions in DNA methylation maintenance and transposon silencing in Arabidopsis. DDM1 has been shown to bind and deposit histone variant H2A.W over transposons for silencing. Additionally, it has the ability to slide nucleosomes, which could provide access to other proteins in the context of heterochromatin. This manuscript presents a structural and biochemical characterization of DDM1 remodeling activity on H2A.W nucleosomes. Osakabe et al demonstrate that DDM1 preferentially slides H2A.W nucleosomes in an ATP-dependent manner. Structural characterization via CryoEM shows that DDM1 primarily binds nucleosomal DNA with minimal interaction with the nucleosome histone core (H4 tail interaction only). However, additional DDM1:nucleosome interactions were identified via crosslinking MS, including unique interactions with H2A.W tail. Mutational analysis demonstrates the importance of both H2W.A and H4 tail in nucleosome sliding assay. Overall, results show that DDM1 has a remodeling activity on H2A.W nucleosomes in addition to the previously known binding and deposition of this variant. While structural analysis is limited based on low resolution of the DDM1:nucleosome complex, the authors conducted insightful experiments via MS-XL and nucleosome sliding assays to support their conclusions. The data provide a strong basis for a remodeling activity of Arabidopsis DDM1. The manuscript is of high technical quality, but insight into DDM1 mechanism and function are somewhat limited. Comments and suggestions to further improve the manuscript are listed below.

Essential Revisions:

1. Figure 3a: Please state the local resolution range of DDM1 and nucleosome core in the text. Based on Supp Fig 2e, the density for DDM1 is 5-6+ Å. At this resolution, if you cannot confidently identify the position of amino acid side chains, they should not be shown in the figure. Additionally, this figure would be improved by including the density of the H4 tail, since histone tails are historically hard to identify due to their flexibility. As Snf2 is not in this figure, it is confusing to include the residue labels.
2. Please comment on the observation that DDM1 is primarily bound to the nucleosomal DNA and that there is no observed contact with histones, specifically H2A.W (i.e., state that the C-terminal tail is disordered in structure)
3. Figure 4d: WT nucleosome values are lower than previous panels (both H2A and H2A.W)?

a. H2A mutant “substantially enhanced” is ~12%, which is still less than 50% of H2A.W shown in previous panels at ~30% (still less than H2A.W in this panel too). What differences could account for this? this should be explained

b. Recommended to soften the statement that this is an “essential role” as activity is still observed at 10% without it.

c. Line 164 “DDM1 binds the H2A.W nucleosome through interactions with the specific H2A.W C-terminal residues” Binding assays with these mutants would strengthen this claim.

4. Supplemental Figure 5: Did MS verify the interactions with the H4 tail that were seen in the structure? Comment on why there are many observed crosslinks with histones and the structure only shows interaction with DNA?

5. An undiscussed topic that would add to the discussion is other histone variants and PTMs found in Arabidopsis. Jamge et al, 2023 found that H3 variants (H3.1/H3.3) form heterotypic nucleosomes and do not associate with a specific H2A variant. How do you anticipate this to effect DDM1 activity and the proposed model? Furthermore, they report that DDM1 uses the same conserved sites to bind both H2A.W and H2A.Z. Bourguet et al, 2022 found that H2A.W cooperates with H3 lysine 9 dimethylation. Expanding this topic in the discussion will place the new DDM1 mechanistic insight into the larger context of chromatin dynamics.

Additional Minor Comments:

1. For the nucleosome sliding assay, the author used terms such as “drastically higher” or “substantially enhanced.” These conclusions would be strengthened with a statistical analysis for significance.

2. Line 122 “The N-terminal tail of H4 is located near the ATPase core domain of DDM1.”

a. This statement would be strengthened by including distances in Figure 3a. As show, DDM1 residues are red residues and H4 is blue, this is misleading with the electrostatic potential scale in the same figure panel. What is charge of the H4 tail residues? Can you see more of this tail compared to the nucleosome alone structure? Adding this discussion could strengthen claim that the H4 tail is bound in the DDM1 acidic pocket.

1. Figure 2b- what pdb is used for free nucleosome, specifically is it H2A or H2A.W?

2. Please include in the manuscript text if the entire DDM1 was used in the structure.

3. Figure 3b: What is the 80 bp band? Is there an explanation for its disappearance in the H2A.W nucleosome sample only?

4. Figure 3a: I think residue 557 should be a D, based on the sequence in Supp Fig 3.

5. Figure 4: What is the distance for each contact and the estimated length of the dashed line? Is it reasonable for the C-terminal tail to reach that far based on amino acid length? If possible, the dashed lines should be in the same position for each orientation and connect to the residue (yellow circle) (example: contact #3 in bottom right extends past the yellow circle)

6. Figure 4c: The authors note that the specific bands corresponding to DDM1:nucleosome complexes disappear in nucleosome lacking the N-term H4 tail, but there is still a clear shift? Why is there more of this band for H2A than H2A.W?

7. Line 145 “Two residues (K203 and K208) of DDM1 crosslinked to the C-terminal tail of H2A.W were close to the regions interacting with nucleotides in the complex Snf12/nucleosomes.”
 - a. Without more context, this sentence doesn't make sense. If you want to make a comparison to another structure, more explanation is required.

8. Fig 5: Where is the density of DDM1 in these figures? Indicate where DDM1 is sitting on DNA (arrow or asterisk if density isn't shown)

9. Figures 5d and S4d and e, could be a new supplemental figure to maintain the order in manuscript text

10. Please include in the figure legend that the nucleosome remodeling assay is normalized to -DDM1

Reviewer #2 (Remarks to the Author):

I co-reviewed this manuscript with one of the reviewers who provided the listed reports as part of the Nature Communications initiative to facilitate training in peer review and appropriate recognition for co-

reviewers.

Reviewer #3 (Remarks to the Author):

Osakabe, Takizawa et al. report the structure of an H2A.W bound nucleosome and the structure of the *A. thaliana* chromatin remodeler DDM1 bound to the H2A.W nucleosome. DDM1 binds the nucleosomal substrate as observed for other chromatin remodelers. Additionally, the authors employ biochemical assays to show that DDM1 is able to shift nucleosomes and that nucleosome shifting by DDM1 is stimulated by the C-terminal tail of H2A.W. The presented work is mostly descriptive and contains limited amounts of additional mechanistic insight as the biochemical analysis remains very limited (cf. major comments). The cryo-EM data analysis is solid, although some important aspects to judge data quality are missing from the current manuscript (cf. minor comments).

Major comments

1. The present nucleosome shift assay is sufficient to demonstrate that DDM1 is able to shift nucleosomes and measure the effects of the C-terminal tail of the H2A variants/tailless H4. However, this assay is not state of the art to show nucleosome sliding as the read-out of the nucleosome shift is only indirect. The authors should repeat the assay and use an actual nucleosome sliding assay where the moved nucleosomes are directly observed through a shift on a NativePAGE gel. Otherwise, the authors cannot exclude that the observed generation of DNA truncations through the restriction digest is not due to secondary effects where the ATP simply induces a conformation shift in DDM1 that favors restriction digest.

2. The nucleosome shift assay is only giving very limited read-outs in regard to the mechanistic effect that the H2A.W tail has on DDM1 activity. The authors should additionally perform experiments that demonstrate if the H2A.W tail impacts ATPase hydrolysis rates or if the H2A.W tail only impacts the coupling of ATPase hydrolysis and DNA translocation. Same is true for the shift assay in the H2A and H2A.W comparison. Does the H2A.W nucleosome stimulate ATPase activity of DDM1 or is it simply easier to remodel H2A.W nucleosomes by achieving higher coupling rates? The authors should address these questions experimentally.

3. Much of the model that is presented in Fig. 6 is in no way explained by the data presented in the manuscript. If the authors include a model figure, they should focus solely on a model figure that directly pertains to their findings.

4. The authors show unwrapping of DNA in the presence of DDM1. It remains unclear if this is only a cryo-EM artifact or is indeed directly related to DDM1. Therefore, the authors should test via a biochemical/biophysical assay if binding of DDM1 to the nucleosome induces nucleosome unwrapping.

Minor comments

1. This reviewer disagrees with the adjective used in line 104 “drastically”. There is no doubt that H2A.W nucleosomes are more easily remodelled, but it seems to only be a modest effect (about 3X stimulation).

2. The authors should give precise numbers of how many rounds of 2D classification were performed (Classification trees).

3. The scale bar for 2D classes is missing for Supplementary Figure 2 and 7.

4. Map-To-Model FSC curves are missing.

5. 3D FSC plots are missing for both cryo-EM reconstructions.

6. Map-To-Densities figures are missing. These figures should show that key interactions (e.g., DNA distortion, H4 tail interaction etc.) are correctly modelled.

7. The authors should point out that there is competing work from the Martienssen lab (doi: 10.1101/2023.07.11.548598)

Reviewer #4 (Remarks to the Author):

In this manuscript, the authors study a chromatin remodeling factor Snf2 homolog in plant, named DDM1 (DECREASE IN DNA METHYLATION 1). DDM1 is known for deposit H2A.W variant in Arabidopsis. They determined cryo-EM structures of H2A.W nucleosome with and without bound DDM1, respectively at resolutions of 2.9Å and 4.7Å. The DDM1-bound nucleosome is significantly more disordered including only 111 base pairs of histone-bound DNA, instead of 145 base pairs. In the low-resolution DDM1-bound nucleosome structure, the authors observed interactions between H4 N-terminal basic residues and DDM1, but the interactions between the C-terminal H2A.W and DDM1 is absent, probably due to the

disordered nature of the structure. They used crosslinking mass spectrometry to establish the DDM1-H2A.W interactions. Furthermore, the authors used mutagenesis of histones, N-terminal deletion mutant of H4 and swapping the C-terminal tails of H2A and H2A.W, and nucleosome sliding assays to conform the structural observations.

Overall, this is a well characterized study. One implication of the DDM1-induced flexibility is to increase the nucleosome accessibility of DNA binding proteins, including DNA methyltransferases. I will suggest that the authors perform DNA methylation assay using the nucleosome substrates in the presence and absence of DDM1.

In the last paragraph of Discussion, the authors suggested a similarity between Arabidopsis DDM1 and mammalian HELLS, in guiding DNA methylation. Is this similarity reflected in the amino acid sequence similarity between DDM1 and HELLS? If so, the sequence of HELLS should be included in the alignment of DDM1 and Snf2 shown in Figure S3. If not, the speculation should be made clear that DDM1 and HELLS do not share sequence similarity.

REVIEWER COMMENTS

Reviewer #1 (Remarks to the Author):

DDM1 is an Snf2-type chromatin remodeler that functions in DNA methylation maintenance and transposon silencing in Arabidopsis. DDM1 has been shown to bind and deposit histone variant H2A.W over transposons for silencing. Additionally, it has the ability to slide nucleosomes, which could provide access to other proteins in the context of heterochromatin. This manuscript presents a structural and biochemical characterization of DDM1 remodeling activity on H2A.W nucleosomes. Osakabe et al demonstrate that DDM1 preferentially slides H2A.W nucleosomes in an ATP-dependent manner. Structural characterization via CryoEM shows that DDM1 primarily binds nucleosomal DNA with minimal interaction with the nucleosome histone core (H4 tail interaction only). However, additional DDM1:nucleosome interactions were identified via crosslinking MS, including unique interactions with H2A.W tail. Mutational analysis demonstrates the importance of both H2W.A and H4 tail in nucleosome sliding assay. Overall, results show that DDM1 has a remodeling activity on H2A.W nucleosomes in addition to the previously known binding and deposition of this variant. While structural analysis is limited based on low resolution of the DDM1:nucleosome complex, the authors conducted insightful experiments via MS-XL and nucleosome sliding assays to support their conclusions. The data provide a strong basis for a remodeling activity of Arabidopsis DDM1. The manuscript is of high technical quality, but insight into DDM1 mechanism and function are somewhat limited. Comments and suggestions to further improve the manuscript are listed below.

Essential Revisions:

Comment 1)

1. Figure 3a: Please state the local resolution range of DDM1 and nucleosome core in the text. Based on Supp Fig 2e, the density for DDM1 is 5-6+ Å. At this resolution, if you cannot confidently identify the position of amino acid side chains, they should not be shown in the figure. Additionally, this figure would be improved by including the density of the H4 tail, since histone tails are historically hard to identify due to their flexibility. As Snf2 is not in this figure, it is confusing to include the residue labels.

Reply)

We thank the reviewer for this comment. In the revised manuscript, we described the local resolution of DDM1 and the nucleosome core in the text (p.6, ll.20-24). As

Reviewer #1 suggested, we removed the position of the amino acid side chain due to the low resolution of the DDM1-H2A.W nucleosome complex structure. We also showed the H4 tail densities in the H2A.W nucleosome and DDM1-H2A.W nucleosome complex for the comparison of the H4 tail structures, and removed the residue labels of Snf2 to avoid confusion. These figures are now shown in Fig. 7c and Supplementary Fig. 13 of the revised manuscript.

Comment 2)

2. Please comment on the observation that DDM1 is primarily bound to the nucleosomal DNA and that there is no observed contact with histones, specifically H2A.W (i.e., state that the C-terminal tail is disordered in structure)

Reply)

We appreciate Reviewer #1's request to clarify our observations in the cryo-EM structure of the DDM1-H2A.W nucleosome complex. We described the disordered structure of the C-terminal tail of H2A.W and the lack of interactions between DDM1 and H2A.W in the cryo-EM structure of the DDM1-H2A.W nucleosome complex in the text (p.9, ll.2-5).

Comment 3)

3. Figure 4d: WT nucleosome values are lower than previous panels (both H2A and H2A.W)?

Reply)

In the process of the revision that Reviewer #3 suggested, we performed the assay to directly detect the nucleosome sliding on the DNA. We then found that DDM1 slides on nucleosomes containing H2A and H2A.W with the same efficiency. The previous nucleosome sliding assay with restriction enzymes may have detected the lower flexibility of the entry/exit nucleosomal DNA specifically occurring in the H2A.W nucleosome, but not the H2A nucleosome. Therefore, we removed all results regarding the nucleosome sliding assay with restriction enzymes in the revised manuscript. The new results of the "REAL nucleosome sliding assay" are presented in Fig. 7 of the revised manuscript.

Comment 3a)

a. H2A mutant "substantially enhanced" is ~12%, which is still less than 50% of

H2A.W shown in previous panels at ~30% (still less than H2A.W in this panel too).
What differences could account for this? this should be explained

Reply)

Again, as explained above, we removed all descriptions regarding the results of nucleosome sliding assay with the restriction enzyme.

Comment 3b)

b. Recommended to soften the statement that this is an “essential role” as activity is still observed at 10% without it.

Reply)

We removed this statement accordingly.

Comment 3c)

c. Line 164 “DDM1 binds the H2A.W nucleosome through interactions with the specific H2A.W C-terminal residues” Binding assays with these mutants would strengthen this claim.

Reply)

We removed the results of the nucleosome sliding assay with mutant nucleosomes containing C-terminal tail-swapped H2A variants in the revised manuscript. However, we observed the potential interaction between DDM1 and the H2A.W C-terminal tail by crosslinking mass spectrometry, as shown in Fig. 6 and Supplementary Fig. 10 of the revised manuscript. We then confirmed that these interactions are structurally possible in the DDM1-H2A.W nucleosome complex. We stated this point in the text (p.9, ll.8-22).

Comment 4)

4. Supplemental Figure 5: Did MS verify the interactions with the H4 tail that were seen in the structure? Comment on why there are many observed crosslinks with histones and the structure only shows interaction with DNA?

Reply)

We thank the reviewer for this comment. Indeed, our crosslinking mass spectrometric analyses did not detect the interaction between the N-terminal tail of H4 and the acidic

pocket of DDM1, in contrast to our observations of the cryo-EM structure of the DDM1-H2A.W nucleosome complex. We reason that the acidic pocket of DDM1 is enriched with acidic residues and devoid of lysine residues, and therefore we could not detect the crosslinking by DSS-H12/D12. This is mentioned in the text (p.12, ll.18-20). Most of the crosslinking interactions are observed between the disordered tail regions of the histone and DDM1. The disordered histone tails are not visible in the cryo-EM technique. This is the reason why many observed crosslinks (most of invisible histone tails) cannot be visualized by cryo-EM. It should be noted that the crosslinking interactions in the substructural classes, which are discarded during the three-dimensional reconstruction, may also be detected in our crosslinking experiments. In the revised manuscript, we described the possible interaction area of the nucleosomal H2A.W C-terminal tail, and revealed that the entire DDM1 region can be interact with it. This result is presented in the new Fig. 6b. We removed the crosslinking mass spectrometry results for the DDM1-H2A nucleosome because we did not obtain the cryo-EM structure of the DDM1-H2A nucleosome complex. Finally, we detected two possible crosslinks between DDM1 and H2A.W, as shown in Fig. 6, and one crosslink between DDM1 and H2B, as shown in Supplementary Fig. 10.

Comment 5)

5. An undiscussed topic that would add to the discussion is other histone variants and PTMs found in Arabidopsis. Jamge et al, 2023 found that H3 variants (H3.1/H3.3) form heterotypic nucleosomes and do not associate with a specific H2A variant. How do you anticipate this to effect DDM1 activity and the proposed model? Furthermore, they report that DDM1 uses the same conserved sites to bind both H2A.W and H2A.Z. Bourguet et al, 2022 found that H2A.W cooperates with H3 lysine 9 dimethylation. Expanding this topic in the discussion will place the new DDM1 mechanistic insight into the larger context of chromatin dynamics.

Reply)

This is an important issue and we thank Reviewer #1 for raising it. Our cryo-EM structure of the DDM1-H2A.W nucleosome complex and the current nucleosome sliding assay results indicate that DDM1 requires only the N-terminal tail of H4, and does not require H2A variants to slide nucleosomes. As a previous study indicated that H4 tail acetylation weakens the interaction with DDM1 (Lee *et al.*, 2023, *Cell*), at least the PTMs of H4 contribute to the nucleosome sliding activity of DDM1. This result was supported by our current findings, showing that the nucleosome sliding activity of

DDM1 was decreased by the removal of the N-terminal tail of H4, as shown in Fig. 7d and e in the revised manuscript. In our previous biochemical study, the same regions of DDM1 bound both H2A.W and H2A.Z in a pull-down assay using histone dimers, but not nucleosomes. In contrast, we did not observe the interaction between DDM1 and H2A variants within nucleosomes in our cryo-EM structure of the DDM1-H2A.W nucleosome complex. These results suggest that we could separate the function of DDM1 into two roles: 1) the deposition of H2A.W onto transposons, and 2) the sliding of nucleosomes regardless of H2A variants. In addition, our crosslinking mass spectrometric analyses identified the interaction between the N-terminal tail of H3 and the C-terminal tail of H2A.W. This observation may be important to understand the link between H2A.W and H3 lysine 9 dimethylation for transposon silencing, which would contribute to a thorough discussion regarding the mechanistic insights about the establishment or maintenance of repressive epigenetic marks over transposons. We discussed our observation of the interaction between the N-terminal tail of H3 and the C-terminal tail of H2A.W and its possible role in transposon silencing in the revised manuscript (p.12, l.21- p.13, l.4).

Additional Minor Comments:

1. For the nucleosome sliding assay, the author used terms such as “drastically higher” or “substantially enhanced.” These conclusions would be strengthened with a statistical analysis for significance.

Reply)

As we removed the results of nucleosome sliding assay with restriction enzyme, we did not use such phrases pointed by Reviewer #1 in the revised manuscript.

2. Line 122 “The N-terminal tail of H4 is located near the ATPase core domain of DDM1.”

a. This statement would be strengthened by including distances in Figure 3a. As show, DDM1 residues are red residues and H4 is blue, this is misleading with the electrostatic potential scale in the same figure panel. What is charge of the H4 tail residues? Can you see more of this tail compared to the nucleosome alone structure? Adding this discussion could strengthen claim that the H4 tail is bound in the DDM1 acidic pocket.

Reply)

We thank Reviewer #1 for this constructive comment. First, we measured and showed the distance of the C α atoms between the residue in the H4 tail and its possible binding residue in DDM1, because the resolution of our DDM1-nucleosome cryo-EM structure is 4.7 Å, which is insufficient for identifying the side chains. These new data are shown in Supplementary Fig. 13b. Second, we changed the colors of H4 and DDM1 to avoid confusion with the electrostatic potential scale, and showed the electrostatic potential scales of H4 tail and DDM1 in a different panel, Fig. 7c. Third, we presented the structure of the DDM1-free nucleosome next to the DDM1-nucleosome complex to emphasize our observation that the N-terminal tail of H4 in the nucleosome alone was disordered, but detected by an interaction with DDM1's acidic pocket. These new data are also shown in Fig. 7c.

1. Figure 2b- what pdb is used for free nucleosome, specifically is it H2A or H2A.W?

Reply)

We used the H2A.W nucleosome for the comparison of the nucleosomal DNA structure around SHL-2. We mentioned the use of the H2A.W nucleosome in Fig. 2b and its legend.

2. Please include in the manuscript text if the entire DDM1 was used in the structure.

Reply)

As Reviewer #1 suggested, we mentioned that the entire/full length DDM1 was used for the cryo-EM structure in the text (p.6, ll.16-19).

3. Figure 3b: What is the 80 bp band? Is there an explanation for its disappearance in the H2A.W nucleosome sample only?

Reply)

The 80 bp band indicates the product after cleavage by Bsh1236I, which recognizes the sequence close to the dyad axis of the nucleosomal DNA, meaning that this enzyme cleaved nucleosome-free DNA. However, as we removed the nucleosome sliding assay with restriction enzymes, the previous Figure 3b is not shown in the revised manuscript.

4. Figure 3a: I think residue 557 should be a D, based on the sequence in Supp Fig 3.

Reply)

Since the Asp 557 residue is located far away from the Arg19 residue of the H4 N-terminal tail detected in our cryo-EM structure, we removed our discussion about this residue in the revised manuscript, as shown in Supplementary Fig. 13b.

5. Figure 4: What is the distance for each contact and the estimated length of the dashed line? Is it reasonable for the C-terminal tail to reach that far based on amino acid length? If possible, the dashed lines should be in the same position for each orientation and connect to the residue (yellow circle) (example: contact #3 in bottom right extends past the yellow circle)

Reply)

We thank Reviewer #1 for this critical comment. We carefully checked the possible interaction between DDM1 and the C-terminal tail of H2A.W by making the 115.35 Å radius corresponding to residues 113-140 of H2A.W (the central point is the C α atom of His113 of H2A.W), which indicates the possible crosslinking area of the H2A.W Lys140 by DSS-H12/D12. Our new approach revealed that the Lys140 and Lys147 residues of H2A.W could contact the Lys208 and Lys342 residues of DDM1, respectively. These new results are shown in Fig. 6 of the revised manuscript.

6. Figure 4c: The authors note that the specific bands corresponding to DDM1:nucleosome complexes disappear in nucleosome lacking the N-term H4 tail, but there is still a clear shift? Why is there more of this band for H2A than H2A.W?

Reply)

We supposed that the observation of multiple bands of the DDM1-nucleosome complex might reflect the various binding modes of DDM1 to nucleosomes as shown in Supplementary Fig. 6c, in which we detected extra densities in addition to the nucleosome by 3D classification. We prepared new nucleosomes with and without the N-terminal tails of the H4 nucleosome and performed the electrophoresis mobility shift assay. Our current results indicated no clear binding differences between H2A.W nucleosomes with or without this tail, as shown in Supplementary Fig. 14. Therefore, we removed the description about the disappearance of the band corresponding to the DDM1-nucleosome complex to avoid confusion.

7. Line 145 “Two residues (K203 and K208) of DDM1 crosslinked to the C-terminal

tail of H2A.W were close to the regions interacting with nucleotides in the complex Snf12/nucleosomes.”

a. Without more context, this sentence doesn't make sense. If you want to make a comparison to another structure, more explanation is required.

Reply)

In the first version of the submitted manuscript, we intended to present that the C-terminal tail of H2A.W might upregulate the ATPase activity of DDM1, such that DDM1 specifically slides the H2A.W nucleosome. However, as we have now described throughout our responses, our current biochemical results suggested that the ATPase activity of DDM1 with H2A.W was similar to that with H2A, and DDM1 slides nucleosomes containing both H2A and H2A.W with almost the same efficiency, as shown in Fig. 7 and Supplementary Fig. 12. Therefore, we removed this description to avoid confusion.

8. Fig 5: Where is the density of DDM1 in these figures? Indicate where DDM1 is sitting on DNA (arrow or asterisk if density isn't shown)

Reply)

In Fig. 3 of the revised manuscript, we show the cryo-EM density of DDM1 in the DDM1-nucleosome complex to clarify where DDM1 binds nucleosomes, for the structural comparison of the entry/exit nucleosomal DNA ends between DDM1-bound and DDM1-free nucleosomes.

9. Figures 5d and S4d and e, could be a new supplemental figure to maintain the order in manuscript text

Reply)

We moved these figures to the supplementary figures accordingly. In the revised manuscript, the previous Fig. 5d and Supplementary Fig. 4d and 4e are now shown as Supplementary Figs. 16, 15a, and 15b, respectively.

10. Please include in the figure legend that the nucleosome remodeling assay is normalized to -DDM1

Reply)

We included the description about the normalization of the nucleosome remodeling assay in the figure legend of Fig. 7.

Reviewer #2 (Remarks to the Author):

I co-reviewed this manuscript with one of the reviewers who provided the listed reports as part of the Nature Communications initiative to facilitate training in peer review and appropriate recognition for co-reviewers.

We thank Reviewer #2 for the contribution to peer review of our manuscript. We hope that we addressed all of your concerns in the revised manuscript.

Reviewer #3 (Remarks to the Author):

Osakabe, Takizawa et al. report the structure of an H2A.W bound nucleosome and the structure of the *A. thaliana* chromatin remodeler DDM1 bound to the H2A.W nucleosome. DDM1 binds the nucleosomal substrate as observed for other chromatin remodelers. Additionally, the authors employ biochemical assays to show that DDM1 is able to shift nucleosomes and that nucleosome shifting by DDM1 is stimulated by the C-terminal tail of H2A.W. The presented work is mostly descriptive and contains limited amounts of additional mechanistic insight as the biochemical analysis remains very limited (cf. major comments). The cryo-EM data analysis is solid, although some important aspects to judge data quality are missing from the current manuscript (cf. minor comments).

Major comments

Comment 1)

1. The present nucleosome shift assay is sufficient to demonstrate that DDM1 is able to shift nucleosomes and measure the effects of the C-terminal tail of the H2A variants/tailless H4. However, this assay is not state of the art to show nucleosome sliding as the read-out of the nucleosome shift is only indirect. The authors should repeat the assay and use an actual nucleosome sliding assay where the moved nucleosomes are directly observed through a shift on a NativePAGE gel. Otherwise, the authors cannot exclude that the observed generation of DNA truncations through the restriction digest is not due to secondary effects where the ATP simply induces a conformation shift in DDM1 that favors restriction digest.

Reply)

We really appreciate this comment of Reviewer #3 for revision. We set up and performed the new experiments to monitor the nucleosome sliding activity of DDM1 without a restriction enzyme. As a result, we found that our previous observation detected the lower flexibility of the entry/exit nucleosomal DNA ends of H2A.W, but we interpreted this result as the H2A.W-specific nucleosome sliding activity. This inconsistent result might be because our previous results reflected the low flexibility of the entry/exit nucleosomal DNA ends of the H2A.W nucleosome compared to the H2A nucleosome. In the revised manuscript, we removed all results of the nucleosome remodeling assay with restriction enzymes. Instead, we performed the actual nucleosome sliding assay as this reviewer suggested, and found that DDM1 slides nucleosomes containing both H2A and H2A.W with the same efficiency. These new results are now shown as Fig. 7 in the revised manuscript.

Comment 2)

2. The nucleosome shift assay is only giving very limited read-outs in regard to the mechanistic effect that the H2A.W tail has on DDM1 activity. The authors should additionally perform experiments that demonstrate if the H2A.W tail impacts ATPase hydrolysis rates or if the H2A.W tail only impacts the coupling of ATPase hydrolysis and DNA translocation. Same is true for the shift assay in the H2A and H2A.W comparison. Does the H2A.W nucleosome stimulate ATPase activity of DDM1 or is it simply easier to remodel H2A.W nucleosomes by achieving higher coupling rates? The authors should address these questions experimentally.

Reply)

We thank Reviewer #3 for this critical comment. We performed the ATPase assay with nucleosomes containing H2A and H2A.W. Consistent with our previous observation (Osakabe *et al.*, 2021, *Nat. Cell Biol.*), DDM1 showed the ATPase activity with DNA compared to DDM1 alone. In addition, we observed higher ATPase activity with nucleosomes rather than DNA, and this activity was detected with both the H2A and H2A.W nucleosomes, as shown in Supplementary Fig. 12 of the revised manuscript. These results also support our current observation of the same nucleosome sliding efficiency of DDM1 for the nucleosomes containing H2A and H2A.W, as shown in Fig. 7a and b of the revised manuscript.

Comment 3)

3. Much of the model that is presented in Fig. 6 is in no way explained by the data

presented in the manuscript. If the authors include a model figure, they should focus solely on a model figure that directly pertains to their findings.

Reply)

We changed the model based on our results, as shown in Fig. 8 of the revised manuscript.

4. The authors show unwrapping of DNA in the presence of DDM1. It remains unclear if this is only a cryo-EM artifact or is indeed directly related to DDM1. Therefore, the authors should test via a biochemical/biophysical assay if binding of DDM1 to the nucleosome induces nucleosome unwrapping.

Reply)

We thank the reviewer for this critical comment. We performed the restriction enzyme susceptibility assay and FRET assay to investigate if the flexibility of the entry/exit nucleosomal DNA ends is increased by DDM1 in solution. Our restriction enzyme susceptibility assay results suggested that DDM1 increased the DNA unwrapping from the histone octamer at the entry/exit regions of the nucleosome without disassembly of nucleosome, as shown in Fig. 4 of the revised manuscript. Furthermore, our FRET assay showed the same trends as our observations with the restriction enzyme susceptibility assay, as shown in Fig. 5.

Minor comments

1. This reviewer disagrees with the adjective used in line 104 “drastically”. There is no doubt that H2A.W nucleosomes are more easily remodelled, but it seems to only be a modest effect (about 3X stimulation).

Reply)

Since we did not observe the H2A.W-specific nucleosome sliding activity with the new experiments, as shown in Fig. 7, we removed this description.

2. The authors should give precise numbers of how many rounds of 2D classification were performed (Classification trees).

Reply)

We added the numbers of 2D classification rounds, as shown in Supplementary Figs. 2, 3, and 6 of the revised manuscript.

3. The scale bar for 2D classes is missing for Supplementary Figure 2 and 7.

Reply)

We added the scale bar for the 2D classes shown in Supplementary Figs. 2, 3, and 6 of the revised manuscript.

4. Map-To-Model FSC curves are missing.

Reply)

We added the Map-To-Model FSC curves, as shown in Supplementary Figs. 2, 3, and 6 of the revised manuscript.

5. 3D FSC plots are missing for both cryo-EM reconstructions.

Reply)

We added the 3D FSC plots, as shown in Supplementary Figs. 2, 3, and 6 of the revised manuscript.

6. Map-To-Densities figures are missing. These figures should show that key interactions (e.g., DNA distortion, H4 tail interaction etc.) are correctly modelled.

Reply)

We showed the Map-To-Densities figures for the interaction between H2A.W and H3, the DNA distortion, and the interaction between the H4 tail and DDM1, as shown in Supplementary Figs. 4, 7, and 13 of the revised manuscript, respectively.

7. The authors should point out that there is competing work from the Martienssen lab (doi: 10.1101/2023.07.11.548598)

Reply)

We mentioned and described the structural differences between the work from the Martienssen lab, the Du lab, and our current study in the text with Supplementary Fig.

17. Since the PDB file published by the Du lab is not available yet, we used the structure of the DDM1-nucleosome complex (PDB ID: 7UX9) for the comparison. We demonstrated that their observations for the binding mode of DDM1 to nucleosomes were almost the same as our results, while we found different structures for the loop of DDM1 facing toward H3. Intriguingly, the flexible features of the entry/exit nucleosomal DNA ends were observed only in our cryo-EM structure. We reason that this discrepancy might come from the use of histone H4 from different species (Martienssen lab used *Xenopus* H4) for the nucleosome reconstitution. It should be noted that there are two amino acid differences between *Xenopus* H4 and *Arabidopsis* H4, and these substitutions are located in the structurally important central helix and the DNA-binding loop. These facts are discussed in the revised manuscript (p.13, 1.16- p.14, 1.6).

Reviewer #4 (Remarks to the Author):

In this manuscript, the authors study a chromatin remodeling factor Snf2 homolog in plant, named DDM1 (DECREASE IN DNA METHYLATION 1). DDM1 is known for deposit H2A.W variant in *Arabidopsis*. They determined cryo-EM structures of H2A.W nucleosome with and without bound DDM1, respectively at resolutions of 2.9Å and 4.7Å. The DDM1-bound nucleosome is significantly more disordered including only 111 base pairs of histone-bound DNA, instead of 145 base pairs. In the low-resolution DDM1-bound nucleosome structure, the authors observed interactions between H4 N-terminal basic residues and DDM1, but the interactions between the C-terminal H2A.W and DDM1 is absent, probably due to the disordered nature of the structure. They used crosslinking mass spectrometry to establish the DDM1-H2A.W interactions. Furthermore, the authors used mutagenesis of histones, N-terminal deletion mutant of H4 and swapping the C-terminal tails of H2A and H2A.W, and nucleosome sliding assays to conform the structural observations.

Comment 1)

Overall, this is a well characterized study. One implication of the DDM1-induced flexibility is to increase the nucleosome accessibility of DNA binding proteins, including DNA methyltransferases. I will suggest that the authors perform DNA methylation assay using the nucleosome substrates in the presence and absence of DDM1.

Reply)

We thank Reviewer #4 for their positive comments. We completely agree with the point that the DNA methylation assay using a reconstituted nucleosome would clarify the mechanism by which the nucleosome sliding and unwrapping activity of DDM1 affect the maintenance of DNA methylation. However, it is still difficult to address this point because it requires a nucleosome containing hemi-methylated DNA and some factors including DNA methyltransferases such as MET1, for which preparation methods have not been established yet. Therefore, this point is very important but should be solved as a future issue.

In the last paragraph of Discussion, the authors suggested a similarity between Arabidopsis DDM1 and mammalian HELLS, in guiding DNA methylation. Is this similarity reflected in the amino acid sequence similarity between DDM1 and HELLS? If so, the sequence of HELLS should be included in the alignment of DDM1 and Snf2 shown in Figure S3. If not, the speculation should be made clear that DDM1 and HELLS do not share sequence similarity.

Reply)

We thank Reviewer #4 for this constructive comment. In our previous study (Osakabe *et al.*, 2021, *Nat. Cell Biol.*), we showed that the histone binding regions of DDM1 share conserved regions with mammalian HELLS/LSH. In this study, we identified possible regulatory regions in DDM1, like the AutoN domain of ISWI. Surprisingly, mammalian HELLS/LSH also showed some sequence similarities with these regulatory regions. We mentioned this point in the discussion of the text with Supplementary Fig. 15 (p.14, ll.7-18).

REVIEWERS' COMMENTS

Reviewer #1 (Remarks to the Author):

Overall, we feel that this manuscript was significantly improved in response to the review process. The new experiments and analyses changed the main conclusions, which highlights that they were necessary before publication. Since the initial submission, the publication of competing work has somewhat limited the novelty of this work. However, we feel that our comments have been adequately addressed and the manuscript can be accepted for publication.

Main Conclusions of Original Manuscript:

1. DDM1 preferentially slides H2A.W
 - o In revision: New sliding assay, now reporting DMM1 slides H2A and H2A.W with the same efficiency
2. H2A.W C-terminal tails and H4 N-terminal tail play important roles in DDM1-mediated nucleosome sliding
 - o In revision: Removed tail swap experiments completely, only show H4-tailless H2A.W
3. DDM1 binds nucleosomal DNA at SHL-2 and SHL+6
 - o In revision: Same conclusions for the DDM1-H2A.W structure
4. DDM1 contacts the H2A.W nucleosome and renders the entry/exit DNA regions flexibly disordered in the H2A.W nucleosome
 - o In revision: Better description of XL-MS data and more analysis of nucleosome flexibility

Main Conclusions of Revision:

1. H2A flexible entry/exit DNA regions compared to H2A.W (not really earth-shattering)
2. DDM1-H2A.W contacts H4 tail and DNA, increases entry/exit DNA flexibility to resemble H2A nucleosome
3. DDMI binds and slides H2A and H2A.W nucleosomes with the same efficiency

Reviewer #2 (Remarks to the Author):

Reviewer #3 (Remarks to the Author):

The authors have addressed by concerns sufficiently, and this reviewer is pleased that the addition of nucleosome sliding assays and ATPase assays could clarify how DDM1 works on nucleosomal substrates.

Reviewer #4 (Remarks to the Author):

The authors addressed partially my concerns. The DNA methylation assay is not feasible at this moment. I support the publication.

REVIEWER COMMENTS

Reviewer #1 (Remarks to the Author):

Overall, we feel that this manuscript was significantly improved in response to the review process. The new experiments and analyses changed the main conclusions, which highlights that they were necessary before publication. Since the initial submission, the publication of competing work has somewhat limited the novelty of this work. However, we feel that our comments have been adequately addressed and the manuscript can be accepted for publication.

Main Conclusions of Original Manuscript:

1. DDM1 preferentially slides H2A.W

o In revision: New sliding assay, now reporting DMM1 slides H2A and H2A.W with the same efficiency

2. H2A.W C-terminal tails and H4 N-terminal tail play important roles in DDM1-mediated nucleosome sliding

o In revision: Removed tail swap experiments completely, only show H4-tailless H2A.W

3. DDM1 binds nucleosomal DNA at SHL-2 and SHL+6

o In revision: Same conclusions for the DDM1-H2A.W structure

4. DDM1 contacts the H2A.W nucleosome and renders the entry/exit DNA regions flexibly disordered in the H2A.W nucleosome

o In revision: Better description of XL-MS data and more analysis of nucleosome flexibility

Main Conclusions of Revision:

1. H2A flexible entry/exit DNA regions compared to H2A.W (not really earth-shattering)

2. DDM1-H2A.W contacts H4 tail and DNA, increases entry/exit DNA flexibility to resemble H2A nucleosome

3. DDMI binds and slides H2A and H2A.W nucleosomes with the same efficiency

Reply)

We thank the reviewer for this comment and summary of our revised manuscript. We are very grateful for this reviewer's previous comments that improved our manuscript.

Reviewer #2 (Remarks to the Author):

I co-reviewed this manuscript with one of the reviewers who provided the listed reports.

This is part of the Nature Communications initiative to facilitate training in peer review and to provide appropriate recognition for Early Career Researchers who co-review manuscripts.

Reply)

We thank this reviewer for reviewing our manuscript.

Reviewer #3 (Remarks to the Author):

The authors have addressed by concerns sufficiently, and this reviewer is pleased that the addition of nucleosome sliding assays and ATPase assays could clarify how DDM1 works on nucleosomal substrates.

Reply)

We thank Reviewer #3 for the critical comments and suggestions. This Reviewer #3 identified the critical point of our misinterpretation regarding the nucleosome sliding activity specific for H2A.W. We really appreciate the comments from Reviewer #3, which greatly improved the revision.

Reviewer #4 (Remarks to the Author):

The authors addressed partially my concerns. The DNA methylation assay is not feasible at this moment. I support the publication.

Reply)

We thank Reviewer #4 for the constructive comments. Comments from Reviewer #4 improved the discussion regarding the conservation of regulatory regions identified in the mammalian HELLS/LSH.